# Formation of three-dimensional bicontinuous structures via molten salt dealloying studied in real-time by in situ synchrotron X-ray nano-tomography

Xiaoyang Liu [1,7], Arthur Ronne[1,7 ✉], Lin-Chieh Yu[1,2], Yang Liu[1,2], Mingyuan Ge[3], Cheng-Hung Lin[1], Bobby Layne[4], Phillip Halstenberg [5,6], Dmitry S. Maltsev [5], Alexander S. Ivanov [6], Stephen Antonelli[3], Sheng Dai [5,6], Wah-Keat Lee[3], Shannon M. Mahurin[6], Anatoly I. Frenkel [1,4], James F. Wishart [4], Xianghui Xiao [3] & Yu-chen Karen Chen-Wiegart [1,3 ✉]

Three-dimensional bicontinuous porous materials formed by dealloying contribute significantly to various applications including catalysis, sensor development and energy storage. This work studies a method of molten salt dealloying via real-time in situ synchrotron three-dimensional X-ray nano-tomography. Quantification of morphological parameters determined that long-range diffusion is the rate-determining step for the dealloying process. The subsequent coarsening rate was primarily surface diffusion controlled, with Rayleigh instability leading to ligament pinch-off and creating isolated bubbles in ligaments, while bulk diffusion leads to a slight densification. Chemical environments characterized by X-ray absorption near edge structure spectroscopic imaging show that molten salt dealloying prevents surface oxidation of the metal. In this work, gaining a fundamental mechanistic understanding of the molten salt dealloying process in forming porous structures provides a nontoxic, tunable dealloying technique and has important implications for molten salt corrosion processes, which is one of the major challenges in molten salt reactors and concentrated solar power plants.

[1] Department of Materials Science and Chemical Engineering, Stony Brook University, Stony Brook, NY, USA. [2] Department of Chemistry, Stony Brook University, Stony Brook, NY, USA. [3] National Synchrotron Light Source II (NSLS-II), Brookhaven National Laboratory, Upton, NY, USA. [4] Chemistry Division, Brookhaven National Laboratory, Upton, NY, USA. [5] Department of Chemistry, University of Tennessee, Knoxville, TN, USA. [6] Chemical Sciences Division, Oak Ridge National Laboratory, Oak Ridge, TN, USA. [7] These authors contributed equally: Xiaoyang Liu, Arthur Ronne. ✉email: arthur.ronne@stonybrook.edu; Karen.Chen-Wiegart@stonybrook.edu

Nanoporous metals created by dealloying methods have intriguing physical–chemical properties and unique morphological evolution that have attracted significant research attention in recent years. Interest has continued to widen as the large surface area and unique three-dimensional (3D) bicontinuous structure of nanoporous materials lends itself well to numerous applications in catalysts, sensors, battery materials, and ultracapacitor materials[1–3]. Research efforts to fundamentally understand the intricate bicontinuous pattern formation[4–9] and its coarsening[10–14] processes have also provided insights revealing the underlying mechanisms. For instance, continuum simulations (e.g. phase-field modeling[8,15,16]) and atomistic simulations including kinetic Monte Carlo simulations[17–21] and molecular dynamics[22,23] model the morphological evolution during both dealloying and coarsening. Simulations also have offered a fundamental understanding of the processing–structure–property relationships such as explaining the atomistic origins on the anomalous compliance[23] and enhanced catalytic properties[17].

Conventionally, aqueous solution dealloying (ASD) has been a prominent dealloying method due to its simplicity. An acid is used as the dealloying agent to selectively remove a component from a parent alloy; the atoms of the remaining metal(s) rearrange and can form a bicontinuous interconnected structure[24,25]. However, this method is generally limited to creating nanoporous structures in relatively noble metals such as Pd, Pt, Au, and Cu, and the use of acid solution could lead to surface oxidation of some porous metals, as well as creating hazardous wastes. One solution to this is to dealloy parent alloys in a metallic melt, as studied in pioneering work by Harrison et al. in the context of preferential leaching[26]. This liquid metal dealloying (LMD) has been more recently applied to form a range of porous structures including creating porous Ti out of Ti–Cu alloy[27,28], porous Fe/Fe-alloys[29–31] using molten Mg, Ta, and Nb from their Ti-alloys using molten Cu[7,32], and TiVNbMoTa nanoporous high entropy alloys[33]. With this method, the enthalpy of mixing is used to choose a system where the sacrificial metal (B) in a parent alloy (A–B) is miscible with the LMD agent (C), while the remaining metal is immiscible with the dealloying agent. Solid-state interfacial dealloying (SSID) builds on this concept but uses a solid-phase metal dealloying agent rather than the metallic melt. At elevated temperatures, the kinetics are sufficiently fast to enable dealloying to occur, which can be integrated with other solid-state processing methods such as thin-film fabrication[30,34]. However, in both the LMD and SSID cases, the dealloying agent (C) and the dissolved metal (B) form a second interconnected phase (B–C), solidified during the cooling process and intertwined with the target porous metal; consequently, an acid solution is still required to remove this unwanted metal phase from the bicontinuous composites to form a porous metal structure. Therefore, the issues associated with the etching chemical waste and creating surface oxides in the porous materials remain.

The drawbacks of present dealloying methods drive a motivation to explore an alternative, green chemistry method to fabricate porous metal using molten salts. The concept of utilizing the corrosive nature of molten salts at elevated temperatures without an applied potential to dealloy and intentionally create porous structures has not been explored. However, molten halide salt corrosion has been widely studied with a focus on the corrosion of structural alloys such as Ni- and Fe-alloys[35,36]. The corrosion studies on molten salts have shown preferential leaching based upon the redox potential of the metal[37]. Here, simple and abundant salts such as NaCl, KCl, MgCl$_2$, and their mixture can be used to introduce preferential corrosion (dealloying), in contrast to the use of low-temperature ionic liquids in prior studies[38]. These redox potentials are well characterized for chloride molten salts, allowing for predictive design choices of dealloying systems. In addition, molten salts have been applied in electrochemical deoxygenation methods, e.g. FCC Cambridge process to synthesize pure metals such as Ti, Zr, and Ta[39]. These works provide an opportunity to apply molten salts in dealloying to fabricate pure porous metals. As metal oxides can be unstable in molten chloride salts in the presence of trace impurities containing O$_2$ and H$_2$O[37,40], a small amount of oxide formation on the surface of the porous metal may be dissolved away during dealloying. Moreover, once solidified, the molten salt residue in the pores that contain dissolved metals, can simply be washed away to create a porous structure, eliminating the need to introduce an acidic solution for etching.

Generally speaking, molten salt corrosion studies were motivated by the use of molten salts as efficient thermal transfer fluids for large-scale solar concentrated power plants[41], and for molten salt reactors (MSRs) as leading candidates for next-generation nuclear power plants[42,43]. Molten salt corrosion is generally driven by the unavoidable trace amounts of impurities in industrial-scale systems, especially water, and it can be enhanced further by other impurities such as metal ions. While the formation of pores in structural materials due to molten salt corrosion has been widely observed[44,45], the nature of the pore formation and its connection to the deliberate use of dealloying to create nanoporous metals has not been discussed in detail. Compared to other corrosion pathways, such as intergranular corrosion along the grain boundaries, this pore formation process is driven by different mechanisms and it can lead to significantly different morphologies with interconnected porous structures that continue to coarsen at an elevated temperature[11,13]. This implication of forming a porous structure after the corrosion of structural materials in molten salts is critical but has generally been neglected. This tortuous, porous pathway will reduce the effective diffusivity for the dissolving metal ions and trace impurities, thereby altering the corrosion kinetics. Reactive transport processes in the porous system can also directly influence electrochemical behavior and corrosion kinetics[46]. Even in a system with flowing molten salt, a porous structure can still impact the local transport of corrosive species, corrosion products, and fission products. Understanding molten salt corrosion mechanisms in these contexts thus play a significant role in the development of future sustainable energy infrastructures, in addition to enabling a method to create porous metals for functional applications.

In this study, the molten salt dealloying (MSD) method is proposed as an approach to fabricate porous metal structures. This method avoids using harsh etchants which have been widely used in ASD and LMDs and overcomes the oxidation challenge of porous materials. There are also differences in using molten salt instead of molten metal as a dealloying agent. Fundamentally, LMD is based on the difference of mixing enthalpies between elements of the parent alloy and dealloying agent; in contrast, MSD depends on the redox potential difference between the constituents within the parent alloy in the molten salt environment. From the processing perspective, the MSD allows the dealloyed residue components in the pores to be simply washed away and create a porous structure, eliminating the use of etchants as in the LMD method. In addition, the use of low-cost and non-toxic salts, and eliminating the need for acids or liquid metals provide added benefits. To demonstrate the MSD process, a binary Ni–20Cr (weight ratio) alloy was heated at KCl–MgCl$_2$ (50–50 molar ratio) salt mixture at 800 °C, and characterized by in situ synchrotron X-ray nano-tomography. The morphology evolution at the early stage of dealloying and subsequent coarsening processes were visualized in real-time, showing the leaching of Cr as well as bi-continuous structural formation as a result of Ni surface diffusion. With further quantification analysis

on the morphology characteristics, the kinetics of the porous structure formation is discussed.

While a synchrotron-based extended X-ray absorption fine structure (EXAFS) has been used previously to study thermal properties of nanoporous materials[47] and investigate the restructuring of their alloy components in situ[48], the high temperature, sample geometry, and ensemble-averaging nature of X-ray absorption spectroscopy present significant challenges in studying the dealloying processes in molten salts. In this work, the oxidation state of the porous material and its local structure was spatially analyzed by X-ray absorption near edge structure (XANES) spectroscopic imaging with a sub-50 nm spatial resolution[49]. In connection with XANES modeling, the effect of dealloying on the local structure of the porous materials associated with both compositional and bond-length changes was understood. Overall, this work illustrates the feasibility of using dealloying (on the basis of redox potential differences in the molten salt medium) to fabricate 3D bicontinuous, open porous materials. As demonstrated here, this in situ approach permits us to quantify the dealloying and coarsening kinetics associated with alloy corrosion as a function of salt composition and impurity content, which directly connects to the critical issue of understanding how to control corrosion in future molten salt nuclear and solar power plants, both important technologies for future large-scale energy sustainability.

## Results and discussion

**In situ 3D morphology evolution with quantitative analysis.** An in situ X-ray nano-tomography experiment was conducted at the Full-field X-ray Imaging (FXI) beamline at National Synchrotron Light Source (NSLS-II) to directly visualize the dealloying process of Ni–20Cr alloy in molten KCl–MgCl$_2$ at 800 °C (see the "Methods" section for further details). This technique has been demonstrated with a sub-50 nm 3D resolution at NSLS-II[49]. Briefly, Fig. 1a shows the schematic of transmission X-ray microscopy (TXM) and a double-capillary sample design for the experiment; here, an inner quartz capillary containing a microwire (~20 μm in diameter) and filled with salt is encapsulated in an outer quartz capillary. The outer quartz capillary is flame-sealed to avoid exposure to atmospheric oxygen and water in the atmosphere. Photographs taken during the experiment are shown in Fig. 1b. 3D X-ray nano-tomographic data was collected continuously with fly scans while the sample is treated at the selected elevated temperature to induce dealloying. A representative X-ray projection image and a reconstructed external sample volume rendering are shown in Fig. 1c.

Noticeably during the heating in molten KCl–MgCl$_2$ at 800 °C, micro-sized pores formed from dealloying the Ni–20Cr parent alloy, as shown in Fig. 2a and the Supplementary Information (Supplementary Figs. 1 and 2, Supplementary Movies 1–3). Rapid dealloying (corrosion) initiated at the interface of the microwire and molten salt and propagated to the center of the sample, forming an open porous structure. Here, primarily Cr was removed (Cr → Cr$^{2+}$ + 2e$^-$) from the parent alloy Ni–20Cr because the Cr/CrCl$_2$ redox couple has a more negative redox potential than Cr/CrCl$_3$ and Ni/NiCl$_2$ in chloride salts at 800 °C, while studies have shown that CrCl$_3$ may form as well depending on the reaction conditions[37,50]. The cathodic reaction and the overall reactions following the Cr dissolution are much more complex as shown by S. Bell et al. [51]. These reactions involve multiple steps and elucidating them properly would deserve separate studies. After ~16 min at 800 °C, the outer portion of the microwire developed a porous structure, while an inner uncorroded zone remained. The dealloying front (metal–salt interface) continued to progress further to the center of the

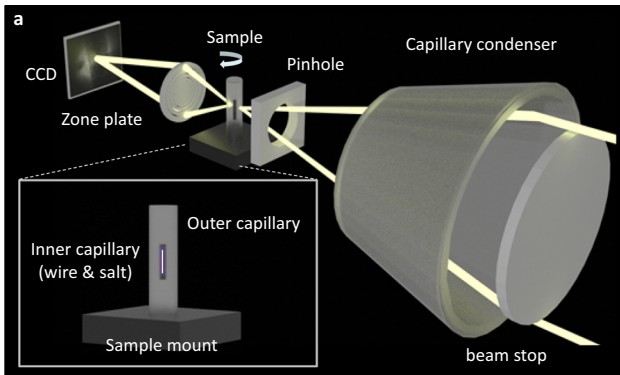

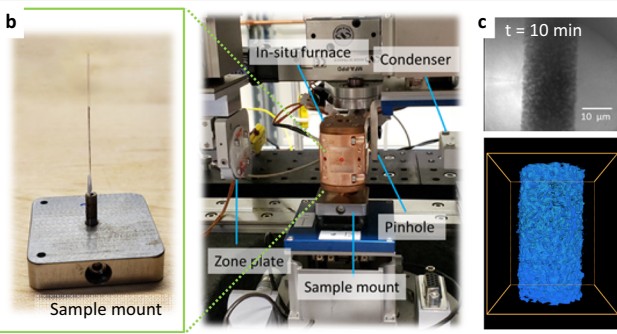

**Fig. 1 In situ X-ray nano-tomography experiment to directly visualize the dealloying process in Ni–20Cr alloy in molten KCl–MgCl$_2$ at 800 °C. a** A schematic showing the optical scheme of transmission X-ray microscopy (TXM) and a double-capillary sample design used for the experiment. **b** Photographs taken at the NSLS-II FXI beamline, 18-ID, NSLS-II showing the experimental setup, and **c** a representative TXM projection image (darker means higher X-ray attenuation) and reconstructed 3D volume rendering (the bounding box dimension is 36.8 × 36.8 × 45.5 μm$^3$) of the microwire after a 10-min reaction, showing the progression of dealloying.

microwire to form a complete 3D porous network of pores by ~33 −47 min. The resulting 3D bicontinuous structure is similar to other porous structures created by various dealloying methods in aqueous solutions and liquid metals[7,27,52,53]. As time progressed, larger pores and ligaments of the porous structure formed due to the prominent coarsening effect at elevated temperature. The reconstructed volumes from the in situ experiment were then quantified to discuss the dealloying and coarsening phenomena and mechanisms in detail.

Firstly, the remaining volume of the solid phase is defined as the pixel counts of the solid phase in the dealloyed structure vs. in the pristine wire. The loss volume of the solid phase is then 1 —the remaining volume of the solid phase as shown in Fig. 2b, the remaining volume of the solid phase decreased initially as the dealloying progressed until it reached ~90% after 46 min. Theoretically, if Cr is completely removed from the Ni–20Cr alloy, the remaining volume of the solid phase would be ~78%. Therefore, some Cr apparently remained in the wire. This is likely due to the increased Cr ion concentration in the salt during dealloying, lowering the driving force for further dealloying. Additional factors that may contribute to the incomplete dealloying include: (1) the exhaustion of the oxidants as the salt was purified and the sample was sealed under an inert environment, and (2) a small amount of residue (typically few at.%) from incomplete percolation dealloying. The discussion of the dealloying mechanism will further validate this observation (vide infra). Figure 2c shows the porosity evolution as a function of the reaction time, where a two-stage process can be clearly observed. (1) During the initial

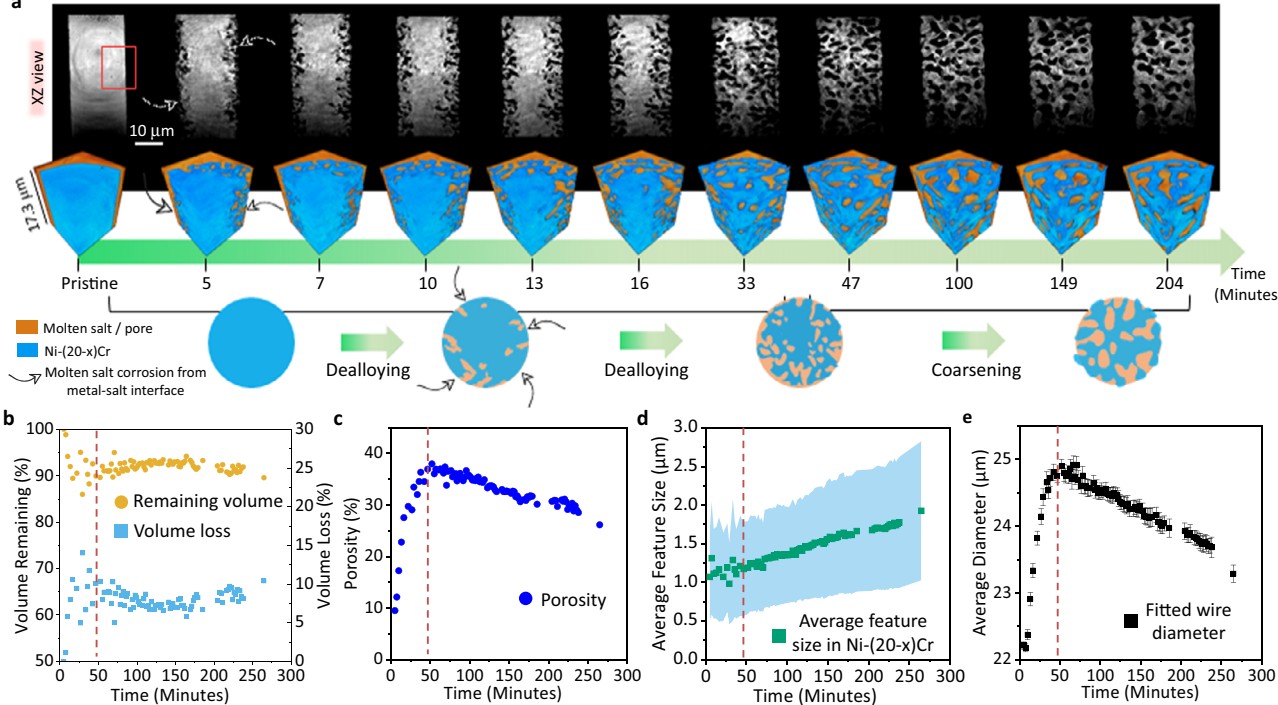

**Fig. 2 In situ synchrotron X-ray nano-tomography on Ni–20Cr reacting with KCl–MgCl₂ molten salt at 800 °C. a** Pseudo-2D cross-sectional view (*XY*) of the full microwire and 3D visualization of a volume of interest (marked with a red rectangle) showing the morphological evolution of the Ni–20Cr microwire. A schematic shows the formation of bicontinuous porous structure during dealloying and the growth of the feature size for pores and ligaments due to coarsening. **b–e** Quantitative analysis of the 3D morphological characteristics including the remaining volume and loss volume of the solid phase, porosity, average ligament feature size, and fitted average wire diameter as a function of reaction time at 800 °C. The blue-shaded range in **d** corresponds to the size distribution of the bicontinuous features. Error bars in **e** indicate the standard deviation from different pseudo-cross-section slices.

stage (time < 47 min) when dealloying was occurring, the porosity increased as Cr was being dissolved. (2) After a fully interconnected porous structure was developed (time > 47 min), the porosity then slightly decreased, which indicated that the dealloying stopped and slight densification occurred; here, the relatively high dealloying temperature can lead to a bulk diffusion, a mechanism responsible for densification as discussed in the later section.

In addition, the average ligament feature size of the formed porous structure (~1.0 μm) did not change significantly during the dealloying stage (Fig. 2d), but only increased slightly as a result of the expected simultaneous coarsening during dealloying. However, at the second stage (time > 47 min) the average feature size gradually increased to ~1.75 μm, consistent with the coarsening effect. Interestingly, the diameter of the porous wire increased significantly when dealloying progressed. This is not seen in other dealloying methods. Considering the thermal expansion, the mean coefficient of thermal expansion ($\bar{a}$) from room temperature to 1000 °F (538 °C) for Ni–20Cr is $8.4 \times 10^{-6}$/°F[54]. The thermal expansion will only lead to an increase of the diameter by ~0.240 μm from the room temperature to 800 °C, which is much smaller than the experimental result. We also noted that a slight sample tilting was observed during the in situ experiment, which has been corrected in the visualization; this slightly tilted angle could influence the diameter quantification but was calculated to be <1 μm. CrCl₃ as a lower X-ray attenuation species was not observed on the surface from the 3D visualizations; in addition, we note that the solubility of CrCl₃ in MgCl₂–KCl at 800 °C has been measured to be 10.9 mol%, and thus the Cr should be able to remain as ionic species in the molten salt mixture without saturation given the current experimental conditions; in other words, we do not expect a growth of CrCl₃ layer contributing to the diameter expansion

Hence, the causes for this diameter expansion still require further investigation.

When dealloying behavior is observed, there are three different major mechanisms to describe these behaviors, as summarized in the work by Chen and Sieradzki[6]: (1) "simultaneous" dissolution and reposition of the more noble element[55], (2) di-vacancy mediated lattice diffusion of the electrochemically active compe-tent to the alloy surface[56] and (3) percolation dissolution[52]. However, these three mechanisms will lead to slightly different morphologies, so they can be discerned qualitatively through microscopy[6]. Dissolution and reposition will generally give a morphology similar to the nodules developed from the clustering of electro-crystallization growth centers while under activation control. Di-vacancy-mediated lattice diffusion spans a range of morphologies from nodules to dendrites, depending on the overvoltage and level of supporting ions in the electrolyte. Finally, percolation dissolution will give a bicontinuous structure when the composition of the alloy is above the percolation threshold, and there is enough surface diffusion of the more noble element (s). Here in our MSD case, a clear bicontinuous structure was observed as shown in Fig. 2a, additionally Ni–20Cr is above the ~18 wt% Cr percolation threshold, supporting the assertion of percolation dealloying for this system[53,57–59].

**Chemical structure study and modeling in MSD.** In addition to the morphology analysis, XANES spectroscopic imaging at the Ni K-edge was performed on the dealloyed Ni–20Cr wire to determine the chemical states and local structural changes of Ni. XANES spectra were analyzed from two selected regions of interests (ROIs) (Fig. 3a, b). ROI 1 was chosen from the center of the wire and ROI 2 from near the surface of the wire. Thus,

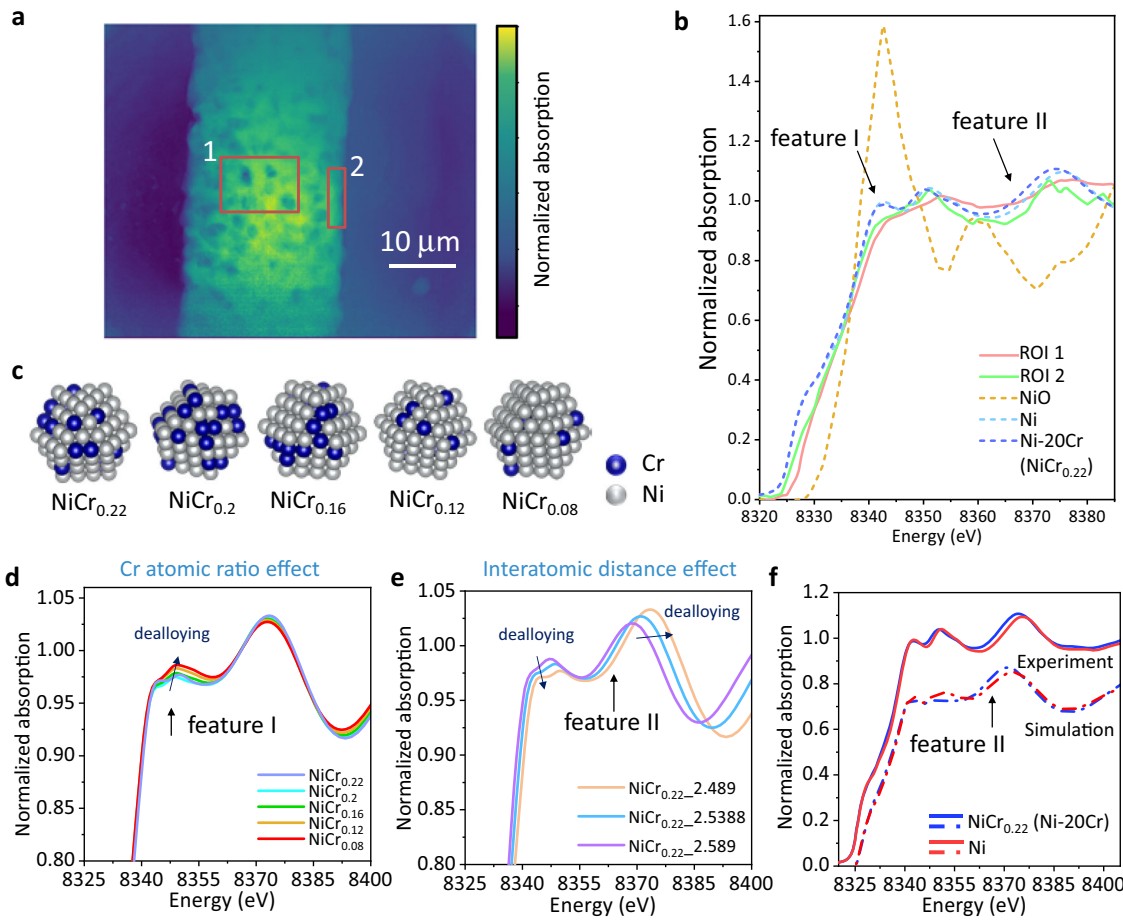

**Fig. 3 XANES spectroscopic imaging and modeling of Ni–20Cr dealloying. a** A representative frame from the 2D XANES spectroscopic mapping; two regions of interests (ROIs) were selected from the porous wire. **b** Normalized Ni K-edge XANES spectra of pristine Ni–20Cr, porous wire after 192.1 min heating at 800 °C [both ROIs as marked in **a**], and the standards (Ni and NiO). **c** Structural representations of six models with different Ni to Cr atomic ratios (78:22, 80:20, 84:16, 88:12, 92:8, and 100:0) and a constant Ni–M (Ni or Cr) distance (2.489 Å). **d** Simulated XANES spectra of the six models with different Ni-to-Cr atomic ratios. **e** Theoretical XANES of the three $NiCr_{0.22}$ (Ni–20Cr) models with different Ni–M (Ni or Cr) distances (2.4890, 2.5388, and 2.5890 Å). When the distance decreased (corresponding to dealloying), the feature at ~8370 eV (feature II) slightly shifted to the higher energy, consistent with the similar shift in the experiment. **f** The XANES spectra comparison between simulated and experimental results of Ni and Ni–20Cr ($NiCr_{0.22}$). The changes in the XANES spectra in **b**, **d**–**f** are shown as feature I and feature II.

ROI 1 would contain contributions from both the surface and interior, while the majority of the contribution would be from the interior region of the sample. In contrast, ROI 2 would contain primarily the contribution from the surface only. Both the averaged XANES spectra showed a similar profile and edge position as the ones from a metallic Ni standard, which indicates that the porous structure was mainly metallic Ni after dealloying. Note that the XANES spectra did not exhibit the characteristic white line feature prominent in NiO, $NiCl_2$, or other $NiCl_xO_y$ compounds[60,61]. The XANES spectrum also does not show a presence of surface oxides on the surface of the dealloyed structure, consistent with our solubility measurements, which show a relatively high solubility (18.5 mol%) of NiO in the $MgCl_2$–KCl melt at 800 °C under an inert atmosphere. In addition, the spectra of the dealloyed sample (both ROIs) showed a decreased in the post-edge feature (feature I), compared with the Ni–20Cr standard. The XANES spectral feature ~8370 eV (feature II) of both ROIs shifted to a higher energy, compared to the Ni–20Cr standard.

To understand the changes in the XANES spectra, simulations were conducted using the FEFF9 code[62]. The effects on the XANES spectral features of two structural parameters were first considered separately to account for dealloying Ni–20Cr into Ni: (1)

Increasing Ni-to-Cr atomic ratio, and (2) decreasing interatomic distance, an effective distance for Ni–M (M = Ni, Cr). Firstly, six models with different Ni-to-Cr atomic ratios (78:22, 80:20, 84:16, 88:12, 92:8, and 100:0) and a constant Ni–M (Ni or Cr) distance (2.489 A) were constructed as shown in Fig. 3c. The atomic ratio of Ni–20Cr is 78:22. The simulated XANES spectra of above models are shown in Fig. 3d. For simulated Ni–Cr alloy, while the atomic ratio of Cr decreased in the models, the intensity of the feature I slightly increased.

Furthermore, change of the Cr atomic ratio in Ni–Cr alloy influences the interatomic distance. Ni metal has an fcc structure (lattice parameter $a = 3.53$ Å) and Cr metal has a bcc structure ($a = 2.91$ Å) with Ni–Ni and Cr–Cr distances of 2.489 and 2.520 Å, respectively. Therefore, in the parent alloy, the effective Ni–M (M = Ni or Cr) distance is longer than that in the pure Ni. Hence, dealloying Ni–20Cr ($NiCr_{0.22}$) will decrease the Ni–M (Ni or Cr) distance in the porous structure. To study the effect of interatomic distance on Ni XANES, three models of Ni–20Cr ($NiCr_{0.22}$) nanoparticles with different Ni–M (Ni or Cr) distances (systematically increased with increments of 2% of the Ni-Ni distance in bulk Ni, 2.489, 2.5388, and 2.589 Å) were constructed. Their simulated XANES spectra are shown in Fig. 3e. When the interatomic distance decreased, the feature II

at ~8370 eV shifted to the higher energy. This is consistent with the experimental result shown in Fig. 3b. Note that when varying the interatomic distance, the intensity of feature I decreased; the combined effect from both the Ni-to-Cr atomic ratio and interatomic distance thus can cancel each other for feature I, leaving it unchanged overall.

Further simulations combined the Cr atomic ratio and interatomic distance effects on the XANES spectra of Ni and Ni–20Cr. The Ni–M (Ni or Cr) distance of 2.507 Å was calculated based on the lattice parameters of Ni–20Cr[63]. As shown in Fig. 3f, both simulated and experimental XANES showed a shift toward a higher energy at ~8370 eV (feature II) from Ni–20Cr to Ni primarily due to the effect of interatomic distance, whereas feature I did not show a difference between Ni and Ni–20Cr spectra, for both simulated and experimental results, consistent with the results shown in Fig. 3c, d. The flattened feature I in the dealloyed sample spectra of ROI 1 and ROI 2 is thus likely caused by the temperature effect (due to the Debye–Waller effect), because the measurement was conducted at an elevated temperature of 800 °C. In summary, theoretical simulation of XANES for Ni–Cr nanoparticles demonstrates the correlation between the Ni-to-Cr atomic ratio, the Ni–M (M = Ni, Cr) interatomic distances, and the changes in the XANES features observed in the experimental data. Overall, with the increase of dealloying time, a Ni-rich phase forms from dealloying Ni–20Cr.

**Kinetics in MSD.** The kinetics of the porous layer thickness growth give insights into the rate-controlling step of MSD, as discussed below. Percolation dealloying in molten salt consists of four steps, analogous to ASD as proposed in prior studies, where the electrolytes are typically acidic aqueous solutions[4,64,65] (Fig. 4a, b): (i) the reactive, oxidizing species in the molten salt diffuses through the pores to access the metal surface; (ii) the more reactive component (Cr) in the parent alloy reacts with the oxidants at the metal–salt interface; (iii) the dissolved Cr ions diffuse out into the molten salt; and (iv) the remaining alloy component (Ni in this case) diffuses along the metal–salt interface to further expose the more reactive (Cr) component from the parent alloy. Here, the analysis focuses on the period of bulk dealloying instead of the very initial dealloying stage. Additional materials parameters may also have impacts on the details of the dealloying process. For instance, the surface roughness may impact the initial period of dealloying related to the porous structure formation which has been studied by high-resolution aberration-corrected scanning transmission electron microscopy on aqueous dealloying of nanostructures of Ag–Au alloys[66]. Future work in this direction by in situ techniques would very beneficial. Here, steps (ii) and (iv) are interface-controlled processes that take place along the pore–solid interface (Fig. 4a), while steps (i) and (iii) are considered to be long-range diffusion processes (Fig. 4b). Note that the anisotropy effect was not considered because the Ni–20Cr wire was polycrystalline, thus

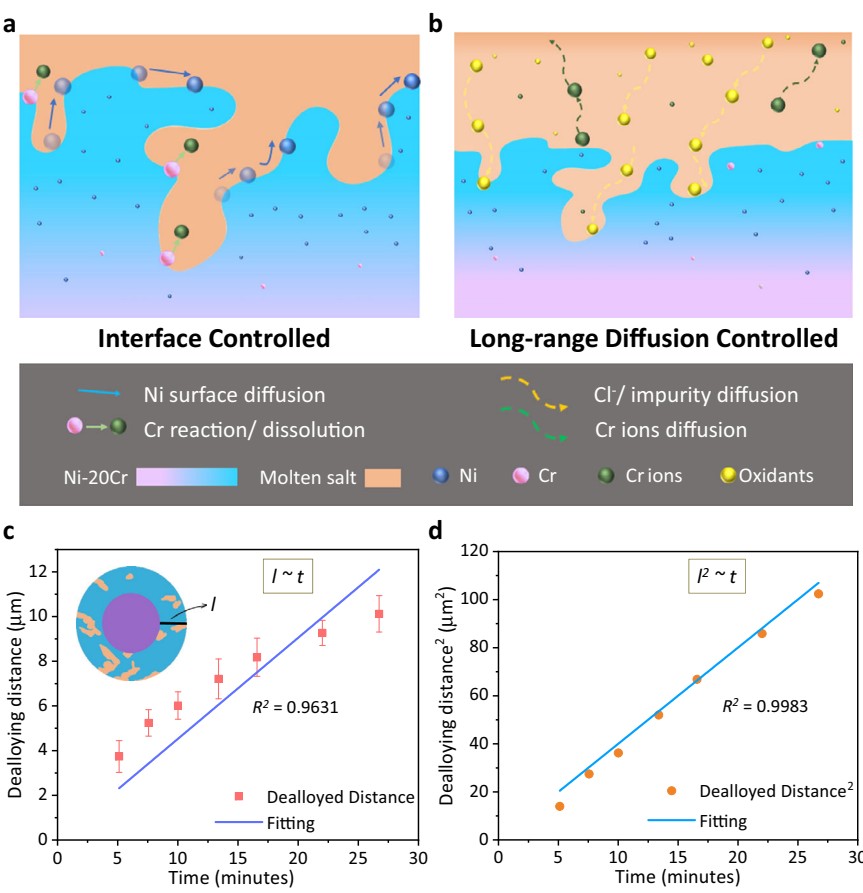

**Fig. 4 Dealloying mechanism in molten salt. a** and **b** Four steps involved in molten salt dealloying of Cr from Ni–20Cr. **a** Interface-controlled processes: Ni surface diffusion, Cr reaction/dissolution, **b** Long-range diffusion-controlled processes: oxidants diffusion and Cr ion diffusion through the tortuous porous structure. **c** and **d** The power-law fitting $l^n \sim t$ of the dealloying distance ($l^n$, $n = 1, 2$) vs. time ($t$) with the goodness of fit. The schematic inset in **c** showed the circle fitting method to determine $l$ using the virtual $XY$ plane images from the in situ X-ray nano-tomography experiment. The error bars in **c** are defined as the standard deviation in calculating l from different pseudo cross-sections. Note that the error bars in **d** are included but they are not visible due to the scale (for the complete data, see Supplementary Table 1).

the average rate and ensemble behavior of all crystallographic orientations are being analyzed. Future work to characterize how different crystallographic orientations may be dealloyed at different rates in the molten salt would be of great interest. Emphasis may be placed on discussing the temperature effect, where the anisotropic effect at the elevated temperature may be less pronounced compared to conventional methods conducted at a room temperature, such as ASD.

Since these four steps need to occur synergistically for the dealloying process to continue, the slower step determines the rate-limiting mechanism of MSD. To determine the rate-limiting mechanism in this system, the dealloying distance ($l$) was measured as a function of dealloying time; this was done by taking the difference of the fitted diameter of the outer wire and the inner non-dealloyed region at each virtual cross-section in the $XY$ plane during the dealloying stage. (Fig. 4c inset).

The power law of $l$ was determined by fitting the $l^n$ as a function of time ($t$) with different $n$ values, corresponding to different mechanisms:

$$l^n \sim t \quad (1)$$

when $n = 1$, a linear relationship between the dealloying time and distance corresponds to an interfacial-controlled process, whereas when $n = 2$, it corresponds to a long-range-controlled diffusion mechanism, where the dealloying rate slows down as a function of dealloying time[8,65], namely:

$$\text{for } n = 1, l \sim t, \frac{dl}{dt} = \text{constant}; \text{for } n = 2, l^2 \sim t, \frac{dl}{dt} \sim \frac{1}{\sqrt{t}} \quad (2)$$

The fitting was constrained to intercept at the origin, corresponding to that at $t = 0$, $l = 0$. The power-law fitting ($n = 1, 2$) of $l^n$ vs. $t$ are shown in Fig. 4c, d. When $n = 1$, the fitting showed a coefficient of determination ($R^2$) of 0.9631, but when $n = 2$, the $R^2$ was 0.9983, the better fit implies that the MSD mechanism under non-flow conditions for up to ~10 μm of the dealloying distance was dominated by long-range-controlled diffusion processes rather than interfacial processes, in other words, step (i) and (iii) as discussed above. Moreover, with the fitted results using $l^2 = kt$, we obtained $k = 3.84115$, thus the time-dependent dealloying rate can thus be determined as

$$\frac{dl}{dt} = \frac{\sqrt{k}}{2} \frac{1}{\sqrt{t}} = 0.9799 \frac{1}{\sqrt{t}} \quad (3)$$

Here we discuss the mechanisms that are involved in these two long-range diffusion steps—namely the inward diffusion of the dealloying agents and the outward diffusion of the dealloyed products through the tortuous pores, both driven by the concentration gradients. For step (i), the supplying of the dealloying agent consists of two motions, the convective transport of the molten salt itself and the diffusion of the corrosive oxidants in the molten salt. The movement of the molten salt into the dealloyed, porous region could be driven by the capillary force. For the diffusion of the corrosive oxidants in the molten salt, although purified salts were used for this study, small amounts (at parts-per-million level) of water ($H_2O$), oxygen ($O_2$), or hydroxide impurities may be contained in the salt because $MgCl_2$ is very hygroscopic[44]. Thus, the diffusion of oxidants in the molten salt could be one of the rate-limiting factors. On the other hand, for step (iv), the long-range diffusion of the Cr ions produced by dealloying at the alloy–salt interface will also alter the dealloying rate according to Nernst equation (Supplementary Information).

An additional comment is that the porous, tortuous diffusion path in the porous metal would decrease the effective diffusion coefficient ($D_{eff}$) of Cr ions away from the dealloying surface. $D_{eff}$ can be related to the diffusion coefficient in a non-porous medium $D_0$ via quantifying the porosity ($\varepsilon$) and the tortuosity ($\tau$) of the structure; here $\tau$ is a 3D geometry parameter, defined as the ratio between a tortuous diffusion path and a straight diffusion path[67,68]

$$D_{eff} = D_0 \frac{\varepsilon}{\tau^2} \quad (4)$$

Interestingly, the nature of the MSD mechanism is a combination of different characteristics of the two other dealloying methods, ASD and LMD. The dealloying reaction mechanism in MSD at the metal–salt interface is driven by electrochemical redox reactions, which is similar to ASD, where the dealloying agents are acid or base solutions; however the reaction kinetics in the case of MSD is more complex because the reaction is driven by small amounts of impurities in the molten salts, rather than a simple acid solution containing a large amount of $H^+$ ions. On the other hand, the ionic diffusion of the dealloying agents and dissolved products in the molten bath is significant in both MSD and LMD case, further highlighting the role of long-range diffusions in determining the dealloying rates. Notably the prior work[7] using a large molten metal bath for LMD also found that the dealloying interface is at or near equilibrium during LMD, and that the rate-limiting step is the liquid-state diffusion of dissolving atoms away from the dealloying interface, also following diffusion-limited kinetics. Note that if the molten salt is supplied in a well-mixed flowing configuration such as in an MSR loop, the system may become interface-controlled, however one would still need to consider the ionic diffusion through porous media.

**Kinetics of coarsening in molten salt.** As evidenced by the increasing of the ligament feature size (Fig. 2d), coarsening dominated after ~47 min of the in situ experiment. To further study the kinetics of the coarsening phenomenon in the molten salt, the power-law relationship between the ligament feature size ($d$) vs. coarsening time ($t$) was determined by fitting the experimental data with the following equation:

$$d^n \sim t \quad (5)$$

where different $n$ values ($n = 1$–$4$) correspond to different mechanisms: $n = 1$ for viscous flow of an amorphous material, $n = 2$ for evaporation and condensation, $n = 3$ for volume diffusion, and $n = 4$ for surface diffusion[69,70]. Coarsening is driven by the overall reduction of the total free energy of a system through reduction of the interfacial area while maintaining a constant volume fraction[71]. All transport mechanisms here are expected to be driven by lowering the surface free energy via transporting the atoms from regions with higher surface curvatures (higher diffusion potentials) to lower surface curvatures (lower diffusion potentials). The fitting was set to intercept at the origin because coarsening would concurrently occur during the dealloying. Here, $n = 2$ corresponds to a potential mechanism where Ni also dissolves slightly into the molten salt, and then redeposits back into the structure. The limited dissolution concentration of $NiCl_2$ in $MgCl_2$–$KCl$–$NaCl$ (44.13/22.79/33.08 mol%) is $1.85 \times 10^{-17}$[72], namely, this is considering the equilibrium between Ni and $NiCl_2$ in a pure chloride melt. While miniscule, this slight solubility allows some amount of Ni to be transported in the melt. Furthermore, our measurements at 800 °C indicate high $NiCl_2$ solubility (45.4 mol%) in the $MgCl_2$–$KCl$ mixture, suggesting that a small amount of the Ni ions from the alloy will be fully diffused in the solution and can potentially be reduced and then deposited back onto the structure. The fitting results for the different mechanisms are shown in Fig. 5a–d; $d^4$ vs. $t$ showed the best fit with an $R^2$ value of 0.9964 which indicated that coarsening in molten salt was dominated by surface diffusion. The scaling

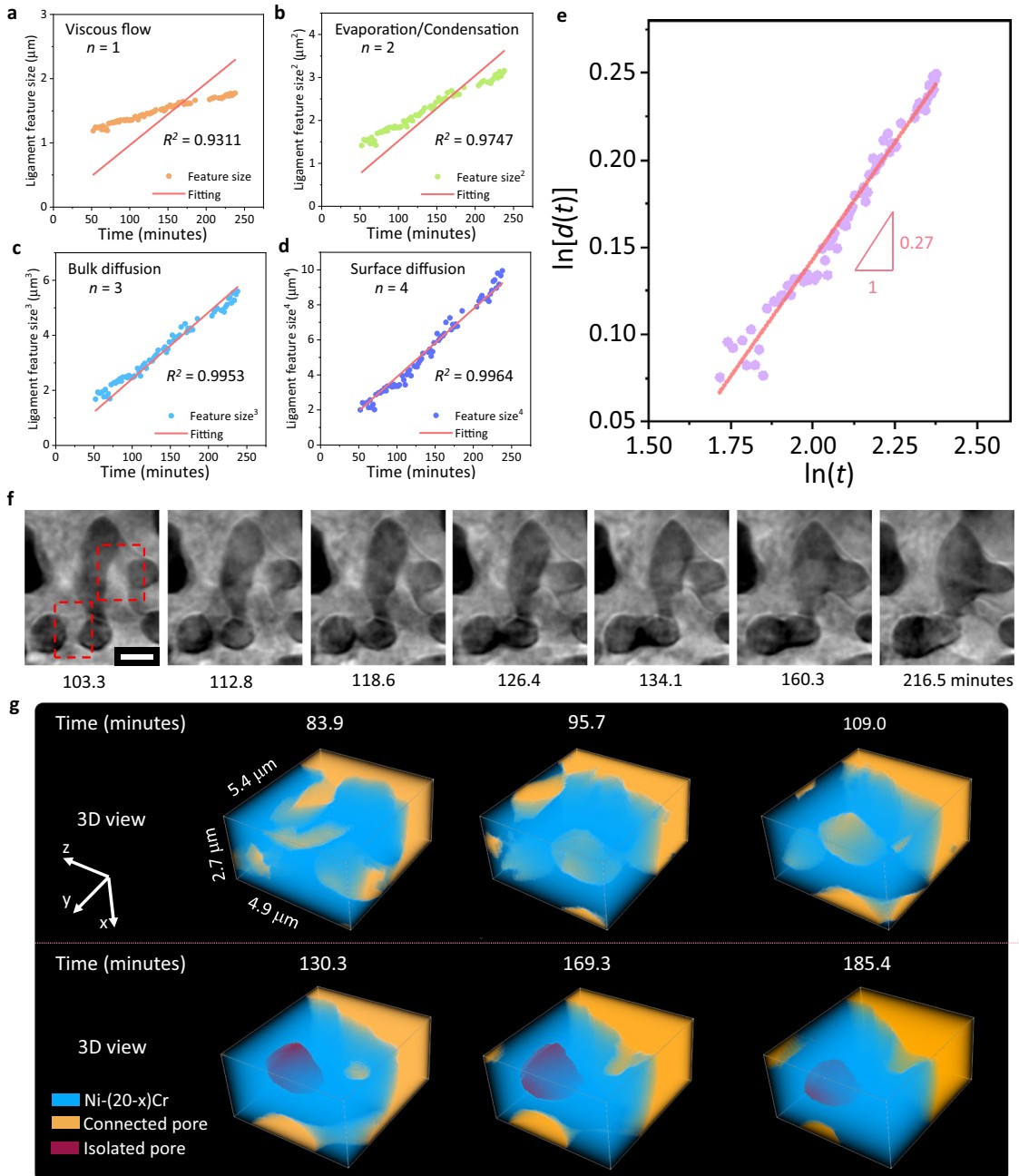

**Fig. 5 Coarsening mechanism power-law fitting for molten salt dealloying (MSD) and a direct observation of enclosed void formation. a–d** Power-law linear fitting ($n = 1$–4) with goodness-of-fit ($R^2$) value of ligament feature size vs. coarsening time after the microwire has been fully dealloyed (~ 47 min during in situ heating), corresponding to different mechanisms: **a** viscous flow, **b** evaporation and condensation (here, Ni dissolution and redeposition), **c** bulk diffusion and **d** surface diffusion. **e** The plot of $\ln[d(t)]$ vs. $\ln(t)$ for measuring the coarsening exponents $n = 1/0.27 = 3.7$ close to 4 indicating that surface diffusion is the primary mechanism for coarsening in molten salt. **f, g** The morphology evolution during the coarsening process at the elevated temperature: **f** ligament pinch-off events; the scale bar is 2 μm. **g** Development of an enclosed pore forming within the ligaments.

exponent from the plot of $\ln[d(t)]$ vs. $\ln(t)$ was determined to be $1/0.27 = 3.7$, close to 4, indicating that surface diffusion is the primary coarsening mechanism (Fig. 5e). The surface diffusion mechanism is consistent with prior studies on coarsening of nanoporous gold by X-ray nano-tomography[13] and recent kinetic Monte Carlo simulations[73], while the coarsening behavior specifically in a molten salt environment could be further studied via simulations, at both continuum and atomic scales. For instance, the model in the study by Weissmüller et al. [73]. showed that the degree of surface faceting or roughness has no apparent effect on the coarsening kinetics; such behavior would be of interest to

investigate in a molten salt environment as the interfacial energy and reactions differ from the conditions in prior studies. Note that $d^3$ vs. $t$ only showed a slightly lower $R^2$ value of 0.9953, bulk diffusion may contribute to the coarsening as well, consistent with the densification observed in Fig. 2c and will be discussed further. The fitting results of $n = 1$ and 2 were poor, meaning that the coarsening in MSD was hardly affected by the viscous flow of the material, or by the dissolving and redepositing of Ni through the molten salt.

Below, we further discuss the surface diffusion mechanism for coarsening since it was determined to be the primary mechanism.

Analogous to the common coarsening in porous materials by the surface diffusion mechanism, coarsening of the porous Ni–(20−x)Cr in the molten salt may proceed by ligament pinch-off and enclosed voids (bubbles) formation controlled by Rayleigh instabilities within the ligaments[11]. Both events were observed here by the in situ X-ray nano-tomography (Fig. 5e, f). First, as discussed in prior coarsening modeling and experiments of porous metals with a bicontinuous structure, in ligament pinch-off events as shown in Fig. 5e, in contrast to a pure surface flattening of concave and convex points, surface diffusion can pull material away from saddle-point curvature ligaments, leading to thinning and breaking-off of ligaments, controlled by Rayleigh instabilities[11,13]. Secondly, considering the formation of enclosed voids (bubbles) within ligaments as shown in Fig. 5f, much work has been done on modeling coarsening in dealloyed metals, as well as ex situ experimental validation of these methods[74,75]; however in situ data is sparse and in situ experimental data of a single pore's evolution is non-existent. Here, the enclosed void formation within a ligament during the MSD was captured by the in situ X-ray nano-tomography from 83.9 to 185.4 min, as shown in Fig. 5c. At first, the pores were connected (orange), however, during the coarsening, one pore (red) then became disconnected from the inter-connected pore phase within the ligament (blue) structure. This void formation within the ligament could be caused by surface-controlled Rayleigh instabilities as well. However, as previously pointed out by Erlebacher[55], the difference between the ligament pinch-off and void formation is that, once an isolated void is formed, it is then separated from any mass transport pathways to continue morphological evolution, whereas a pinched-off ligament could continue to evolve via its connection with the primary structure (Supplementary Movies 4 and 5). Another possible mechanism of the isolated pore formation is that neighboring ligaments collapse accompanied by plasticity at nodes or ligaments, supported by prior atomic simulation models[10]. Recent kinetic Monte Carlo and molecular dynamics simulations also revealed that structures with a low relative density exhibited a surface stress-driven dislocation movement leading to coarsening by ligament collapse[22]. This mechanism was also further supported by observation via in situ transmission electron microscopy in 2D[76].

Note that the porosity was continuously decreasing during the coarsening process (Fig. 2c); however, a sole coarsening effect would maintain a constant porosity. A densification process can occur at the same time due to the relatively high dealloying temperature of 800 °C, which corresponds to a homologous temperature ~0.57 for Ni−xCr (x < 20). Densification has also been observed in other porous metals' coarsening processes at elevated temperatures[13,77]. Surface diffusion can only lead to morphological rearrangement and not a densification process, therefore other mechanisms must be considered here. As shown in Fig. 5a, b, the viscous flow and dissolution/redeposition mechanisms do not correspond well with the experimental data here, and thus the bulk/volumetric diffusion could be primarily responsible for the densification mechanism. By adjusting the dealloying temperature and time, coarsening and densification could be controlled to tune the properties of the porous materials made by MSD method for tailored applications.

To summarize, in this study, a MSD approach was demonstrated to create 3D bicontinuous porous metal by dealloying Ni–20Cr in MgCl$_2$–KCl at an elevated temperature, where Cr is preferably leached in the molten salt bath. In situ synchrotron nano-tomography by TXM was conducted to directly visualize the dealloying process to form an interconnected open porous structure, as well as the subsequent morphological evolution during coarsening. During the dealloying stage, the metal–salt dealloying front progressed rapidly inwards, leading to a volume loss, porosity increase, and sample diameter expansion. Under the prolonged elevated-temperature conditions, the coarsening effect in conjunction with a slight densification became dominant, with ligament feature size and pore size growth and a slight porosity reduction. TXM XANES spectroscopic imaging showed that the porous structure remained metallic Ni, owing to the instability of nickel oxides in chloride molten salts. XANES simulation and the effects of dealloying Ni–20Cr were modeled by considering the increasing Ni-to-Cr atomic ratio and decreasing bond length. The simulation results suitably explained the changes in the XANES spectra features observed experimentally.

The kinetic mechanisms of the dealloying and coarsening processes of MSD were determined from the in situ X-ray nano-tomography analysis. The rate-limiting mechanism in the MSD dealloying of Ni–20Cr without flowing salts was long-range diffusion, namely the diffusion of the oxidant ions and Cr ions in the molten salt. Although the dealloying process relied on electrochemical reactions similar to ASD, the oxidant impurities complicate the dealloying mechanisms in the molten salt. The porous structure decreases the effective diffusivities of the dissolved species, and hence the long-range diffusion could become a rate-limiting step, similar to LMD where molten metals are used as dealloying agents. Furthermore, the coarsening effect was determined to be primarily controlled by surface diffusion, where the ligament pinch-off and enclosed void formation occurred together to form a larger feature size, as observed directly by in situ X-ray nano-tomography. Bulk diffusion also contributed to the coarsening process, causing slight densification and a reduction in porosity of the structure.

MSD eliminates the use of strong acid to create porous metal structures, as in other dealloying methods such as ASD and LMD. Importantly, dealloying in molten salts such as chlorides, preserves the metallic state of the metal when MSD is performed under an inert and dry environment, thus preventing oxide formation. In addition, the heating process could be adjusted to control the coarsening and densification processes, making it possible to tailor the pore size or even to create multi-scale porous structure for a wide range of functional applications. As studying the mechanical properties has been an active research area for the field, future work can also be conducted to study the mechanical behaviors of the porous metals prepared by MSD[9,23,78,79].

This fundamental study on morphological evolution is also directly connected with applications in MSRs and concentrated solar power plants, where corrosion of structural materials in contact with molten salt is a major issue. Corrosion in a porous media that is dynamically evolving is characterized by several interrelated electrochemical, chemical, and physical mechanisms, which are also interconnected to the complex structure of the porous media itself. Future predictive models to consider the reaction rates and materials kinetics for corrosion ideally should consider the detailed porous morphological evolution, as it can directly alter the transport phenomena and reaction kinetics. Future in situ analyses, which can further address a range of chemical and physical parameters that influence both the transport phenomena and chemical/electrochemical reactions as well as their coupled processes are vital for a complete understanding.

## Methods

**Salt purification.** KCl–MgCl$_2$ in a 50:50 molar ratio was selected as the dealloying agent. It is well known that the corrosivity of chloride molten salts is significantly impacted by the presence of oxidizing impurities such as H$_2$O, O$_2$, and HCl[51]. Additionally, oxidizing impurity separation of MgCl$_2$ is challenging as hydrated MgCl$_2$·nH$_2$O undergoes hydrolysis when dehydrated beyond the monohydrate, producing MgOHCl above ~300 °C and MgO above ~550 °C[80,81]. Here MgCl$_2$ was purified via fractional distillation starting from commercial anhydrous salt, taking advantage of the difference in vapor pressure for pure MgCl$_2$, its oxides and hydroxychlorides. The MgCl$_2$ was heated in a distillation column to 925 °C under a constant vacuum of 1 × 10$^{-3}$ Torr. The evaporated, pure MgCl$_2$ then condensed in

the cold zone and contaminants were left in the heated zone. After distillation, dried $MgCl_2$ was then transported into an Ar-filled glovebox without exposure to air and kept in the glovebox for storage. The bulk purified $MgCl_2$ was titrated to determine an oxide concentration of $1.9 \times 10^{-4}$ g of MgO. The KCl was purchased as 99.999% Suprapur reagent grade. The ampule containing KCl was opened inside the glovebox and only handled inside of the glovebox. The KCl and $MgCl_2$ were ground and mixed together with a mortar and pestle in a 50:50 molar ratio.

**Materials composition characterizations**. As-drawn wires, 20 μm in diameter of 80 wt% Ni–20 wt% Cr (99.5% pure, Goodfellow, USA-NI055105) were used for this study. The size and shape were chosen to ensure both allowing sufficient X-ray transmission and adequately fitting within the field of view size of the TXM. As discussed in an earlier paper, the Ni–20Cr wire has been characterized by inductively coupled plasma–optical emission spectrometry (ICP-OES) and ICP-mass spectrometry (MS)[82], included in Supplementary Table 2. The wires were dissolved in a mixture of 8 M $HNO_3$ and 9 M HCl at 150 °C and were initially analyzed using ICP-OES to obtain Ni and Cr content. Higher sensitivity measurements were then performed using ICP-MS.

**TXM sample preparation**. The Ni–20Cr wire was placed into a 0.1 mm diameter, open-ended quartz capillary (Charles Supper). This was then baked out at ~200 °C for at least 3 h to remove any surface water present on the quartz or wire. Inside of an Ar-filled glovebox, in a quartz boat the salt mixture was melted at 550 °C. Using a syringe attached to the end of the 0.1 mm wire-filled capillary, the salt is drawn up into the capillary and immediately solidified; parafilm was used to seal the connection between the syringe and the capillary. This 0.1 mm salt-filled and wire-filled capillary is then placed into a 1.0 mm diameter closed-end capillary, sealed with epoxy and allowed to cure for at least 2 h within the glovebox. The thin inner capillary is used to ensure enough X-ray transmission through the sample, whereas a larger outer capillary is used to ensure a proper sealing of the sample. Then samples are removed from the glovebox and immediately flame-sealed using a miniature benchtop hydrogen torch (Rio Grande) and mounted onto a stainless-steel sample holder using a high-temperature epoxy.

**In situ testing setup and TXM measurements**. In situ X-ray nano-tomography experiments were conducted at the FXI beamline (FXI, 18-ID) at NSLS-II of Brookhaven National Laboratory[49]. The assembled sample of Ni–20Cr wire in 50:50 molar KCl–$MgCl_2$ was mounted onto a TXM sample holder and placed into a miniature furnace integrated with the TXM at FXI, which was jointly developed by NSLS-II and the Molten Salts in Extreme Environments Energy Frontier Research Center[83]. This sample was heated to 800 °C with a ramp rate of 25 °C min, held at 800 °C for 4 h, then cooled to room temperature with a ramp rate of 50 °C min$^{-1}$. The X-ray incident energy used was 8.33 keV, slightly below the Ni K-edge of 8.333 keV for optimal imaging contrast. Images were captured using a lens-coupled CCD detector with $2560 \times 2160$ pixels and a $55.5 \times 46.8$ μm$^2$ field of view. A camera binning of $2 \times 2$ was used, resulting an effective pixel size of 43.34 nm. For each tomography scan an exposure time of 50 ms was used to capture ~1050 projections, with a total acquisition time of 2–2.5 min per scan. The tomography projection images were low-pass filtered before reconstruction with the gridrec algorithm and the reconstructions were done with Tomopy[84,85].

**Visualization and quantification analysis of TXM results**. The 3D tomographic reconstruction stack images were cropped into volumes of 13.0 μm (height) $\times$ 31.2 μm $\times$ 29.9 μm. A batch segmentation of the solid and pore phases was applied based on the thresholding value between the corresponding peaks in the histogram of the 3D images. Visualization of 2D virtual cross-section images and 3D morphology evolution was conducted in commercial software Avizo (Thermo Fisher Scientific, v9.3) on the reconstructed data. The visualization of an isolated pore growth was conducted on segmented images using the software Tomviz[86]. The identification of pore isolation or connection is based on if it has same voxel value in nearest-neighbored voxel.

The remaining volume of the solid phase calculation was determined by the voxel counts of the solid phase in each scan divided by the voxel counts of the solid phase in the pristine sample. To further measure the evolution of diameter, corrosion distance and feature size distribution, two circles were fitted to each scan. An outer circle fitting to determine the outer diameter of the sample at all times was conducted on the XY pseudo-cross sections. The analysis used an algorithm to ensure that the circle enclosed at least 98.8% of the solid phase which was determined by testing different ratio values (Supplementary Fig. 3) Hence, the diameter of the fitted circle was considered to be the diameter of the porous material. Furthermore, another circle within the sample was fitted to determine the dealloying distance. The same origin as used for the outer circle used here; the fitting was conducted to enclose the non-corroded part of the sample within the overall porous structure on the early-stage scans from the in situ experiment, with enclosed materials to be at least 99.4% (Supplementary Figs. 3 and 4). The corrosion distance was the radius difference between above two fitted circles. The best power-law fitting ($n = 2$–$3$) of the corrosion distance in early-stage scans as a function of time was conducted. Finally, the ligament feature size distribution was quantified for the 3D tomographic reconstruction at each of the time point using a Matlab program developed in-house, following established algorithms[87]. An example is shown in

Supplementary Fig. 5 for a dealloying time of 61.0 min. Note that to characterize a bicontinuous structure, it is more appropriate to use a size distribution, instead of a single size value, to represent the feature size in the structure. The average ligament feature size and the standard deviation of the distribution were calculated from each reaction time point, from the cropped images. The standard deviation here does not indicate measurement errors, but rather statistically represents the distribution of the feature size within the structure, owing to the characteristic bicontinuous structure which cannot be represented by a monodisperse system.

**XANES measurements and modeling**. 2D XANES images of the sample were obtained by scanning the Ni K-edge from 8.251 to 8.621 keV at FXI beamline of NSLS-II. The XANES data were processed using PyXAS developed at FXI beamline by Dr. M. Ge and Athena software[88,89]. Standard Ni, Ni–20Cr foil, and NiO XANES were measured at the Beamline for Materials Measurement (BMM) at NSLS-II of BNL in the Ni K-edge energy region using transmission geometry. The Athena software package was used to align and merge multiple scans and perform edge-step normalization of the XANES data.

For XANES modeling, several atomistic coordinates of face-centered cubic structure models of NiCr with different Cr:Ni ratios and Ni–M (Ni or Cr) distances were constructed and simulated by FEFF[62]. The non-structural parameters for XANES simulations were chosen to ensure the best agreement between simulated spectrum for the bulk of Ni foil and the corresponding experimental XANES data. FEFF version 9.6.4 was used for self-consistent calculation within full multiple scattering (FMS) and muffin-tin (MT) approximations. FMS cluster size was chosen at a large value so that the whole cluster was included in the FMS calculations. Random phase approximation (RPA) was used to model core–hole and the default value (2.0 Å) was used for MT radius, as well as complex exchange-correlation Hedin–Lundqvist potential. The structural model of NiCr was based on the perfect cuboctahedrons with 147 atoms.

**Solubility measurements**. Anhydrous $CrCl_3$, $NiCl_2$, and NiO (99.99% purity) were purchased from Aldrich-APL. Salt mixtures for the solubility measurements employing isothermal saturation method were prepared in the glovebox by mixing the purified $MgCl_2$–KCl salt with the respective chromium and nickel chlorides/oxide in the amount exceeding their theoretical solubility limits in the molten $MgCl_2$–KCl at 800 °C. The resulting mixtures were fused at high temperature under vacuum in quartz cells, which were then refilled with ultra-high purity nitrogen to maintain an inert atmosphere. The melt was periodically stirred at 800 °C for 6 h and then allowed to settle for additional 24 h keeping the high temperature regime. The precipitation of undissolved chloride/oxide was visually verified during the experiment. The final probes (two samples for each mixture) were collected from the upper clear part of the melt. After cooling down to room temperature the salt samples were dissolved in 5 wt % nitric acid solution. The potassium, magnesium, and transition metal concentrations were measured by ICP-OES. A schematic diagram of the experimental apparatus used for the solubility measurements is shown in Supplementary Fig. 6.

## Data availability
The datasets generated during and/or analyzed during the current study are available from the corresponding author on reasonable request.

## Code availability
The code used to analyze the X-ray nanotomographic data during the current study are available from the corresponding author on reasonable request.

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

## Acknowledgements

This work was supported as part of the Molten Salts in Extreme Environments (MSEE) Energy Frontier Research Center, funded by the U.S. Department of Energy, Office of Science, Basic Energy Sciences. BNL and ORNL are operated under DOE contracts DE-SC0012704, and DE-AC05-00OR22725, respectively. Work at Stony Brook University was supported by MSEE through a subcontract from BNL. This research used resources, the Full Field X-ray Imaging (FXI, 18-ID) beamline, and the Beamline for Materials Measurement (BMM, 6-BM) of the National Synchrotron Light Source II, a U.S. Department of Energy (DOE) Office of Science User Facility operated for the DOE Office of Science by Brookhaven National Laboratory under Contract DE-SC0012704. Partial support for A. Ronne was provided by an NSF NRT Award in Quantitative Analysis of Dynamic Structures (DGE 1922639) as a fellowship. The authors are grateful to Dr. Bruce Ravel (National Institute of Standards and Technology), scientist at BMM beamline, for his expertise and support of experiments. The authors acknowledged the inputs and support on heater design from Dr. Steve Hulbert (NSLS-II, BNL). Dr. Kazuhiro Iwamatsu is acknowledged for assistance during the sample preparation. Current and former Chen-Wiegart group members are acknowledged for operating the FXI and BMM beamtimes together: Chonghang Zhao, Qingkun Meng, Karol Dyro, Dean Yen, and Brian Conry.

## Author contributions

A.R. and Y.-c.K.C.-W. developed the research idea with inputs from S.M.M. and J.F.W. A.R., M.G., B.L., S.A., J.F.W., X.X., and Y.-c.K.C.-W. discussed and designed the in situ X-ray nano-tomography experiments. P.H. purified the salts under the supervision of S.D. D.S.M. and A.S.I. conducted the solubility measurements. B.L. and A.R. designed the double-capillary in situ sample scheme. A.R., X.L., L.-C.Y. and Y.-c.K.C.-W. conducted the FXI beamtime with the support of X.X., M.G., W.-K.L. and Chen-Wiegart group members. X.L., A.R., and L.-C.Y. conducted data analysis and visualization under the guidance of X.X., M.G., and Y.-c.K.C.-W., with inputs from C.-H.L. XANES modeling was conducted by Y.L. and A.I.F. In situ movies were prepared by X.L. X.L., A.R., and Y.-c.K.C.-W. developed the data interpretation and mechanistic understanding and wrote the manuscript with inputs from all co-authors.

## Competing interests

The authors declare no competing interests.
