## [Peer Review File · Nature Communications]

Reviewers' comments:

Reviewer #1 (Remarks to the Author):

The manuscript is well-structured and easy to read, which allows this reviewer to follow the authors' logic flow. Besides several technical details, the major drawback of this manuscript is that the novelty or extreme importance in the related research field was clearly addressed. As far as this reviewer understand,

- a. The molten salt dealloying method is likely a new approach, but it's significance over the liquid metal dealloying method is not obvious. Thus, scientifically speaking, it may be a derivative than a new.
- b. In situ X-ray nano-tomography experiment appears to be a useful technique, and the reviewer sees the advantage of the in situ approach to kinetics evaluation. However, its spatial resolution may not be sufficient to discuss the detailed kinetics and reaction mechanisms. For example, the average feature size measurement (fig. 2D) suggests that the spatial resolution limit of this techniques is around 1 micro-meter, thus the measured dealloying distance used for dealloying kinetics analysis could have +/- 50% difference. Further, the model seems not taking the surface roughness into account. This would be another factor to change the diffusion length (or reaction surface area) estimation. This reviewer was also wondering whether the crystallographic nature of the surface, i.e., packing density of the contact surface, would change the redox kinetics such as {111} surface dealloyed faster than {001} surface because the Ni-Cr alloy used for this study may not be a single crystal. Shouldn't this anisotropy effect be included in the kinetics modeling?
- c. Although the molten salt dealloying process reduces the surface oxide formation, surface oxidation could happen rapidly once the nanoporous structure was exposed to the atmosphere. In addition, it was not clear if this approach can be free from the potential capillarity effect. Since the authors mentioned "incomplete percolation dealloying" happened at elevated temperature, removing salt especially from the inner part would be challenging.

Minor comments:

- d. For measuring a wire, the spatial resolution of XANES may lead misunderstanding. ROI 1 and 2 (Fig. 3) both contain signals from the surface and the interior with a different ratio. It was not clear how the authors identify the contributions from the surface and the interior, unless the separation was not necessary.
- e. The feature about 8370eV could be from NiCl_x, NiO_x, or NiClO_x. How did the authors eliminate these possibilities?
- f. The Ni-Cr alloy is a solid solution (disordered). Then, it was not clear for this reviewer that how reducing the Cr concentration changes the Ni-Cr distance?

Reviewer #2 (Remarks to the Author):

In this study, a new method of molten salt dealloying (MSD) to realize using the green condition to create specific nanoporous materials. In real-time, in situ synchrotron 3D X-ray nano-tomography provides strong evidence to understand the composition change and kinetic studies during the dealloying process, especially in high-temperature conditions. The article should be accepted with the following minor comments:

1. In this study, the redox potential difference in binary alloy elements was thought as a possible reason to make the selective metal dissolution in binary alloy elements. This could be an interesting and reasonable reason. Any formulations about half-reactions on cathodic and anodic side, respectively, could be provided to make this concept easier to be understandable.
2. In line 145,because the Cr/CrCl₂ redox couple has a more negative redox potential than Ni/NiCl₂ in chloride salts at 800 °C. Could "CrCl₂" mean as the "CrCl₃"?

3. The diameter of the porous wire increased during dealloying progression. Any possible comments to illustrate this interesting behavior.
4. Any additional interesting mechanical properties ever found on this material, eq. the sponge-like elasticity enhancement.
5. The XANES spectrum demonstrates the dealloyed structure composed mainly of metallic Ni owing to an absence of surface oxides on the surface after the dealloying process. Were any NiCl₂ compounds found on the dealloyed structure? A little difference in spectrum between Ni in the dealloyed structure and pure metallic Ni could be related to NiCl₂ but to oxides formation.

Reviewer #3 (Remarks to the Author):

This work reports that corrosion of a metal alloy in a molten salt leads to a bicontinuous solid/pore structure in an extended volume of the original solid. The underlying process is therefore analogous to dealloying, as it has previously been described for corrosion in aqueous media and in molten metal. The manuscript analyzes in detail the evolution of the porosity. In-situ synchrotron x-ray tomography is the key technique, supplemented by x-ray near-edge absorption fine structure analysis. Among the findings are insights into the kinetics and the underlying processes of dealloying and of the subsequent coarsening. These insights are of substantial interest in the fields of dealloying as well as molten-salt corrosion. The character of the study, namely bridging communities (scientists interested in making porous metals and those interested in understanding molten-salt corrosion) adds to its relevance. As its key theme, the manuscript proposes molten-salt dealloying as a new mechanism for forming porous materials. This finding, if new, would indeed be a substantial novelty in the field of dealloying. The formation of extended volume of porous material is also of high interest to the corrosion community. Considering those statements, it would appear that the paper is a very suitable submission to Nature Communications.

There is, however, a prior publication by essentially the same team of authors on essentially the same subject, which anticipates the key novelty. I am referring to the manuscript reference 65, Ronne et al., "Revealing 3D Morphological and Chemical Evolution Mechanisms of Metals in Molten Salt by Multimodal Microscopy", ACS Appl Mater & Interfaces 12 (2020) 17321. Quoting from the abstract of that paper:

"The binary alloy Ni-20Cr developed a bicontinuous porous structure, reassembling functional porous metals manufactured by dealloying. This work elucidates better mechanistic understanding of corrosion in molten salts, which can contribute to the design of more reliable alloys for molten salt applications including next-generation nuclear and solar power plants and opens the possibility of using molten salts to fabricate functional porous materials."

That appears not fundamentally different from the central theme of the present publication. The present publication follows the acting mechanisms to greater depth and thereby does achieve important novel insights which definitely deserve publication. Yet, it would seem that a more specialized journal is suitable for this in-depth follow-up study, rather than Nature Communications with its ambition to report top and qualitatively new insights to a wider audience.

Detailed comments follow.

The manuscript relies overly on the use of abbreviations. This makes the reading arduous and should be corrected.

The discussion of the shift in the Nernst potential addresses graduate-level science that should be well familiar to readers from the dealloying or molten salt corrosion communities; a more concise focus on the novel science might be in order.

The discussion of the state of the art concerning mechanisms for dealloying as well as coarsening

misses the insights from atomistic simulations of dealloying that have been documented in the literature of the past five years.

The data on coarsening kinetics (figure 5) should be displayed in a log-log representation. The discussion of the power-law exponent should include the results of linear regression in that representation. That would provide for an unbiased analysis, strengthening the conclusions.

Throughout the text, the distinction between 1) external sample volume, 2) volume of the solid phase, and 3) volume of the pore phase, along with 4) solid volume fraction needs a dedicated terminology and precise definitions. These should then be consistently adhered to in the entire manuscript, so that the reader can follow the observations and conclusions on those important parameters clearly. As the text stands, this is not warranted.

In several places the paper claims superior friendliness towards the environment. Those passages are not supported by solid arguments. Those arguments should be given (for instance, provide quantifiable and verifiable evidence that dealloying in molten salt is more environmentally friendly than dealloying in acid) or the claims withdrawn.

For this reviewer's personal taste, the passages Acknowledgment and Author Contributions are way too detailed for what readers are interested in. Scientific papers communicate science – letting them evolve into legally watertight documentation of who did or who financed what is not a trend that scientists should support.

Review #1 comments:

The manuscript is well-structured and easy to read, which allows this reviewer to follow the authors' logic flow. Besides several technical details, the major drawback of this manuscript is that the novelty or extreme importance in the related research field was clearly addressed. As far as this reviewer understand,

[Response] We highly appreciate the reviewer for the great efforts and time spent on reviewing the manuscript, as well as the positive notion on the structure of the manuscript. We thank the reviewer also for all the insightful comments. Here we would like to address the comments and questions raised by the reviewer, in hope that the reviewer may reconsider recognizing the novelty and importance of this work.

a. The molten salt dealloying method is likely a new approach, but it's significance over the liquid metal dealloying method is not obvious. Thus, scientifically speaking, it may be a derivative than a new.

[Response] **The molten salt dealloying and liquid metal dealloying (LMD) methods differ significantly both regarding their practical processing aspects, as well as the underlying mechanism associated with the driving force leading to the bicontinuous structure.** Fundamentally, LMD is based on the difference of mixing enthalpies between elements of the parent alloy and dealloying agent. In contrast, MSD depends on the redox potential difference between the constituents within the parent alloy in the molten salt environment. Their mechanisms clearly differ and thus do not support viewing the molten salt dealloying as a derivation of LMD.

From the processing perspective and the significance of the molten salt dealloying over the LMD method - one key issue when using LMD to fabricate a porous structure is the need to remove the 2nd interconnected metallic phase by acid etching, which is not required in the new method of molten salt dealloying presented in this work. This challenge for LMD was stated in the introduction: "However, in both the LMD and SSID cases, the dealloying agent (C) and the dissolved metal (B) form a second interconnected phase (B-C), **solidified during the cooling process and intertwined with the target porous metal; consequently, an acid solution is still required to remove this unwanted metal phase from the bicontinuous composites to form a porous metal structure.**" In contrast, the molten salt dealloying method allows a simpler approach, "once solidified, **the molten salt residue in the pores that contain dissolved metals, can simply be washed away to create a porous structure**, eliminating the need to introduce an acidic solution for etching." Further comparison on the molten salt dealloying methods with the existing aqueous dealloying method and liquid metal dealloying are also elaborated in the results and discussion sections, under '*Kinetics in molten salt dealloying*': *The dealloying reaction mechanism in MSD at the metal-salt interface is driven by electrochemical redox reactions, which is similar to ASD, where the dealloying agents are acid or base solutions; however the reaction kinetics in the case of MSD is more complex because the reaction is driven by small amounts of impurities in the molten salts, rather than a simple acid solution containing a large amount of H⁺ ions. On the other hand, the ionic diffusion of the dealloying agents and dissolved products in the molten bath is significant in both MSD and LMD case, further highlighting the role of long-range diffusions in determining the dealloying rates.*

We have revised the manuscript for further clarification.

[change] Introduction

In this study, the molten salt dealloying (MSD) method, which utilizes the redox potential difference in binary alloy elements, is proposed as a new approach to the fabrication of porous metal structures. This method avoids using harsh etchants which have been widely used in aqueous solution dealloying and liquid metal dealloying methods and overcomes the oxidation challenge of porous materials. There are also differences in using a molten salt instead of a molten metal as a dealloying agent. Fundamentally, liquid metal dealloying is based on the difference of mixing enthalpies between elements of the parent alloy and dealloying agent; in contrast, MSD depends on the redox potential difference between the constituents within the parent alloy in the molten salt environment. From the processing perspective, the molten salt dealloying allows the dealloyed residue components in the pores to be simply washed away and create a porous structure, eliminating the use of etchants as in the liquid metal dealloying method.

b. In situ X-ray nano-tomography experiment appears to be a useful technique, and the reviewer sees the advantage of the in situ approach to kinetics evaluation. However, its spatial resolution may not be sufficient to discuss the detailed kinetics and reaction mechanisms. For example, the average feature size measurement (fig. 2D) suggests that the spatial resolution limit of this techniques is around 1 micro-meter, thus the measured dealloying distance used for dealloying kinetics analysis could have +/- 50% difference. Further, the model seems not taking the surface roughness into account. This would be another factor to change the diffusion length (or reaction surface area) estimation.

[Response] The 1 μm resolution cited by the reviewer is incorrect. The conclusion on suggesting our data could have a $\pm 50\%$ difference is also incorrect. As we mentioned in the method section, the pixel size is 43.34 nm, and the spatial resolution of the transmission X-ray microscope (TXM) in 3D has been demonstrated to be sub-50 nm. The nano-tomography technique by TXM utilizes a Fresnel zone plate as the objective lens to provide magnification and thus provides a much higher resolution than conventional X-ray tomography. The average feature size measurement shown in the Fig. 2D, has a shaded color range of $\pm 50\%$, but this is not the errors from the measurement. Instead, the range represents a distribution of the feature size due to the bicontinuous structure. The average feature size and the standard deviation were calculated from the distribution of the feature size based on the full 3D imaging datasets. Because of the bicontinuous porous structure, the dealloyed features have a size distribution, and are not monodisperse. The ability to conduct such analysis rather demonstrates the advantages of X-ray nano-tomography studying the kinetics because it provides a quantification of morphology in three dimensions. We recognized that the way we presented this figure may have caused confusion and have revised it accordingly.

Importantly, the measured dealloying distance and the corresponding analysis did not have this $\pm 50\%$ difference as the reviewer claimed, because the spatial resolution is on the sub-50 nm scale; this resolution can well resolve the dealloying distance which is on the order of μm , as can be seen in Figure 4. The error bars are marked in Figure 4 for dealloying kinetics analysis and it is well within reason.

We revised the manuscript for clarification.

[Change] *in situ 3D morphology evolution with quantitative analysis*

An *in situ* X-ray nano-tomography experiment was conducted at the Full-field X-ray Imaging (FXI) beamline at National Synchrotron Light Source (NSLS-II) to directly visualize the dealloying process of Ni-20Cr alloy in molten KCl-MgCl₂ at 800 °C (See Methods section for further details). This technique has been demonstrated with a sub-50 nm 3D resolution at NSLS-II.⁴⁹

Method

Finally, the ligament feature size distribution was quantified for the 3D tomographic reconstruction at each of the time point using a Matlab program developed in-house, following established algorithms.⁸⁷ An example is shown in **Figure S 5** for a dealloying time of 61.0 min. Note that to characterize a bicontinuous structure, it is more appropriate to use a size distribution, instead of a single size value, to represent the feature size in the structure. The average ligament feature size and the standard deviation of the distribution were calculated—sizes from time points after 47 minutes were quantified between two circles at the obvious corrosion side on cropped images with the power-law fitting ($n = 1 - 4$) from each reaction time point, from the cropped images. The standard deviation here does not indicate measurement errors, but rather statistically represents the distribution of the feature size within the structure, owing to the characteristic bicontinuous structure which cannot be represented by a monodisperse system.

Supporting Information

Figure S 5 – An example of the ligament feature size distribution characterization of the bicontinuous structure, quantified for the dealloying time of 61.0 min. The distribution was then used to calculate the average feature size (Avg) and the standard deviation (Std), corresponding to the feature size distribution in Figure 2 (D). The average feature size with the distribution range were for each of the reaction time point based on the 3D tomographic reconstruction, and shown in Figure 2 (D).

Figure caption:

Figure 2 – In situ synchrotron X-ray nano-tomography on Ni-20Cr reacting with KCl-MgCl₂ molten salt at 800 °C : ... The blue-shaded range in (D) corresponds to the size distribution of the bicontinuous features. Error bars in (D-E) indicate the standard deviation from different pseudo cross-section slices.

This reviewer was also wondering whether the crystallographic nature of the surface, i.e., packing density of the contact surface, would change the redox kinetics such as {111} surface dealloyed faster

than {001} surface because the Ni-Cr alloy used for this study may not be a single crystal. Shouldn't this anisotropy effect be included in the kinetics modeling?

[Response] We agree with the reviewer that the surface roughness is an important factor to consider for the initial period of dealloying process related to the porous structure formation which has been studied by high-resolution aberration-corrected scanning transmission electron microscopy on aqueous dealloying of nanostructures¹, which showed that “The difference in the roughness evolution between the nanorod and the nanocube suggests a link between the initial period and the population of surface defects.... “ It is important to note that, in this cited literature where these two nanostructures with different surface roughness evolution were analyzed, it showed “In both cases, surface dealloying is followed by surface diffusion and eventually exposes sufficient sites for bulk dealloying”. **Since we are analyzing the kinetics during the period of ‘bulk dealloying’ instead of the very initial dealloying stage, the notion on the importance of the surface roughness does not contradict to the key findings of our work,** namely the emphasis on revealing if the bulk dealloying behavior in molten salt is rate-limited by long-range diffusion or surface phenomena (surface diffusion or interfacial reactions). In our revised manuscript, we included the discussion of this important *in situ* work in the field, as well as discern more clearly the kinetics that we aim to quantify.

For the anisotropic effect, indeed it would be very interesting to study how different crystallographic orientation may be dealloyed at different rates. However, since our work is studying a polycrystalline structure (as the reviewer also pointed out), the average rate and ensemble behavior of all crystallographic orientations are being analyzed here. Note that this is consistent with other work in the field that studied the kinetics during dealloying, including the two *Nature Communications* papers^{2, 3} – neither kinetics model considered the anisotropic effect. We surely agree that this would indeed be a very interesting aspect to study in follow-up work and to be included in future models and have included this comment in the revised manuscript. We should mention that no noticeable behavior of faceting during dealloying or coarsening was observed. As molten salt dealloying is conducted at an elevated temperature, the surface energy difference between different crystallographic planes is expected to be less than at room temperature, and thus the anisotropic effect may be less pronounced compared to conventional methods conducted at room temperature, such as aqueous dealloying.

There is no doubt that there can be many other materials factors which influence dealloying processes, and it is not possible to discuss all factors in one manuscript. We sincerely hope that our work is a pioneer in the field that could invite many follow-up studies for further investigation.

[Change] *Kinetics in molten salt dealloying (MSD)*

Here, the analysis focuses on the period of bulk dealloying instead of the very initial dealloying stage. Additional materials parameters may also have impacts on the details of the dealloying process. For instance, the surface roughness may impact the initial period of dealloying related to the porous structure formation which has been studied by high-resolution aberration-corrected scanning transmission electron microscopy on aqueous dealloying of nanostructures of Ag-Au alloys.⁶⁶ Future work in this direction by *in situ* techniques would very beneficial.

Here, steps (ii) and (iv) are interface-controlled processes that take place along the pore–solid interface (Figure 4A), while steps (i) and (iii) are considered to be long-range diffusion processes (Figure 4B). Note that the anisotropy effect was not considered because the Ni-20Cr wire was polycrystalline, thus the average rate and ensemble behavior of all crystallographic orientations are being analyzed. Future work to characterize how different crystallographic orientations may be dealloyed at different rates in the molten salt would be of great interest. Emphasis may be placed on discussing the temperature effect, where the

anisotropic effect at the elevated temperature may be less pronounced compared to conventional methods conducted at a room temperature, such as aqueous solution dealloying.

c. Although the molten salt dealloying process reduces the surface oxide formation, surface oxidation could happen rapidly once the nanoporous structure was exposed to the atmosphere. In addition, it was not clear if this approach can be free from the potential capillarity effect. Since the authors mentioned “incomplete percolation dealloying” happened at elevated temperature, removing salt especially from the inner part would be challenging.

[Response] We agree with the reviewer’s opinion that some porous materials will form surface oxides when it is exposed in the air, but this is an intrinsic property of the metals, independent of the fabrication methods. For metals that are not easily oxidized in air, but may be oxidized in acids, molten salt dealloying method provides an alternative to avoid or at least reduce the surface oxidation formation since it is free from using the acids. The key is that, as a processing method itself, molten salt dealloying does not further promote surface oxide formation during the processing.

The potential capillary effect and the removal of the salts from the porous structure should not be of concern. E.g. aqueous dealloying has been carried out on samples with several mm thickness through sub-10 nm pores, and etching residual metallic phase using acid from the composites fabricated by liquid metal dealloying has been done on bulk samples, also through pore size similar to the ones in our work.

Minor comments:

d. For measuring a wire, the spatial resolution of XANES may lead misunderstanding. ROI 1 and 2 (Fig. 3) both contain signals from the surface and the interior with a different ratio. It was not clear how the authors identify the contributions from the surface and the interior, unless the separation was not necessary.

[Response] As shown in Figure 3A, ROI 1 was chosen from the center of the wire and ROI 2 from near the surface of the wire. Thus, ROI 1 would contain contributions from both the surface and interior, while the majority of the contribution would be from the interior region of the sample. In contrast, ROI 2 would contain primarily the contribution from the surface only. As explained in the manuscript, “Both the averaged XANES spectra collected from the central bulk region of the wire [Figure 3 A-B, region of interest 1 (ROI 1)] and from the region near the surface (ROI 2) showed a similar profile and edge position as the ones from a metallic Ni standard, which indicates that the porous structure was mainly metallic Ni after dealloying.” We made changes for further clarification.

[Changes] *Chemical structure study and modeling in molten salt dealloying*

XANES spectra were analyzed from two selected regions of interests (ROIs) (Figure 3 A-B). ROI 1 was chosen from the center of the wire and ROI 2 from near the surface of the wire. Thus, ROI 1 would contain contributions from both the surface and interior, while the majority of the contribution would be from the interior region of the sample. In contrast, ROI 2 would contain primarily the contribution from the surface only. Both the averaged XANES spectra collected from the central bulk region of the wire [Figure 3 A-B, region of interest 1 (ROI 1)] and from the region near the surface (ROI 2) showed a similar profile and edge position as the ones from a metallic Ni standard, which indicates that the porous structure was mainly metallic Ni after dealloying.

e. The feature about 8370eV could be from NiCl_x, NiO_x, or NiClO_x. How did the authors eliminate these possibilities?

[Response] Our XANES results show that this feature at ~ 8370 eV from the sample is significantly different from the NiO standard XANES (Figure 3 B). Furthermore, the XANES spectrum of the sample did not exhibit the characteristic white line feature (the strong peak right after the edge) in the NiO standards. The absence of the white line strongly supports that the Ni phase in our sample was metallic Ni and not NiO. Similarly, for NiCl_x and NiCl_xO_y, a clear white line feature would also be present according to the literature (See ref.⁴ for NiCl_x, also shown below in Figure R1 and ref.⁵ for NiCl_xO_y), which was not observed in the XANES spectrum of our sample. Changes were made to the manuscript to reflect this clarification.

Figure R1 – XANES spectra of the NiO, ROI 1 and ROI 2 from the manuscript (shown in Figure 3B) and the NiCl₂ spectra from ref 5. The white line features are highlighted by red arrows in both figures for NiO and NiCl₂ which are different from the spectra of ROI 1 and ROI 2.

[Change] *XANES measurements and modeling*

Both the averaged XANES spectra collected from the central bulk region of the wire [Figure 3 A-B, region of interest 1 (ROI 1)] and from the region near the surface (ROI 2) showed a similar profile and edge position as the ones from a metallic Ni standard, which indicates that the porous structure was mainly metallic Ni after dealloying. Note that the XANES spectra did not exhibit the characteristic white line feature prominent in NiO, NiCl₂ or other NiCl_xO_y compounds.^{60, 61}

f. The Ni-Cr alloy is a solid solution (disordered). Then, it was not clear for this reviewer that how reducing the Cr concentration changes the Ni-Cr distance?

[Response] It is correct that Ni-20Cr alloy at this composition is a solid solution. We would like to clarify that the distance used in our modeling was specified as the interatomic distance (an ‘effective distance’

considering Ni-M, M = Ni or Cr, not only the Ni-Cr distance as claimed by the reviewer: As written in our manuscript, “Therefore, in the parent alloy, the effective Ni-M (M = Ni or Cr) distance is longer than that in the pure Ni. Hence, dealloying Ni-20Cr (NiCr0.22) will decrease the Ni-M (Ni or Cr) distance in the porous structure. To study the effect of interatomic distance on Ni XANES, ...”. This change in interatomic distance has also been reported in the literature, measured by neutron scattering, see reference Jin *et al.*⁶, Figure 5(a), also shown below in Figure R2. We made further clarification on this point to avoid confusion.

Figure R2 – Neutron diffraction data showed the difference in lattice parameter between Ni and Ni-20Cr, from ref 7.

[Change] *Chemical structure study and modeling in molten salt dealloying*

The effects on the XANES spectral features of two structural parameters were first considered separately to account for dealloying Ni-20Cr into Ni: 1) Increasing Ni-to-Cr atomic ratio, and 2) decreasing interatomic distance, **an effective distance for Ni-M (M = Ni, Cr).**

XANES measurements and modeling

For XANES modeling, several atomistic coordinates of face-centered cubic structure models of NiCr with different Cr:Ni ratios and Ni-M **(Ni or Cr)** distances were constructed and simulated by FEFF⁴⁹.

Review #2 minor comments:

In this study, a new method of molten salt dealloying (MSD) to realize using the green condition to create specific nanoporous materials. In real-time, in situ synchrotron 3D X-ray nano-tomography provides strong evidence to understand the composition change and kinetic studies during the dealloying process, especially in high-temperature conditions. The article should be accepted with the following minor comments:

[Response] We are very grateful for the very positive comments from the reviewer. Please find our responses to your comments below.

1. In this study, the redox potential difference in binary alloy elements was thought as a possible reason to make the selective metal dissolution in binary alloy elements. This could be an interesting and reasonable reason. Any formulations about half-reactions on cathodic and anodic side, respectively, could be provided to make this concept easier to be understandable.

[Response] Because the focus of the work was not on the chemical reactions, we did not include the reactions in the manuscript. We added the possible anodic reaction in the revised manuscript, namely the dissolution of Cr ($\text{Cr} \rightarrow \text{Cr}^{2+} + 2e^-$). The cathodic reaction, and the overall reactions following the Cr dissolution are much more complex as shown by S. Bell *et al.*⁷. These reactions involve multiple steps and elucidating them properly would deserve separate studies. We included this notion in the revised manuscript.

[Change] *In situ 3D morphology evolution with quantitative analysis*

Rapid dealloying (corrosion) initiated at the interface of the microwire and molten salt and propagated to the center of the sample, forming an open porous structure. Here, primarily Cr was removed ($\text{Cr} \rightarrow \text{Cr}^{2+} + 2e^-$) from the parent alloy Ni-20Cr... The cathodic reaction, and the overall reactions following the Cr dissolution are much more complex as shown by S. Bell *et al.*⁵¹. These reactions involve multiple steps and elucidating them properly would deserve separate studies.

2. In line 145,because the Cr/CrCl₂ redox couple has a more negative redox potential than Ni/NiCl₂ in chloride salts at 800 °C. Could “CrCl₂” mean as the “CrCl₃”?

[Response] Based on the literature,⁸ the redox potential of Cr/CrCl₂ is more negative than both Cr/CrCl₃ and Ni/NiCl₂ at an elevated temperature. However, the reviewer is correct that we have also seen in our recent work (K. Bawane *et al.*, submitted) that the final corrosion product is Cr³⁺ instead of Cr²⁺. We clarified this point in the revised manuscript.

[Change] *In situ 3D morphology evolution with quantitative analysis*

... because the Cr/CrCl₂ redox couple has a more negative redox potential than Cr/CrCl₃ and Ni/NiCl₂ in chloride salts at 800 °C, while studies have shown that CrCl₃ may form as well depending on the reaction conditions.^{37, 50}

3. The diameter of the porous wire increased during dealloying progression. Any possible comments to illustrate this interesting behavior.

[Response] This is an interesting phenomenon which has not been seen in other dealloying methods. We have considered the thermal expansion and tilting angle of the sample during experiment. However, these two possible reasons may account for the increasing of diameter by only less than 1 μm . In addition, a growth layer of CrCl_3 was considered. However, there is no evidence of Cr agglomeration on the surface in the TXM images, which would be shown by a lower attenuating species, nor is this expected based upon our measured solubility of CrCl_3 . Thus, we believe further investigation is needed to understand the expansion of the wire diameter, as explained in the manuscript. We may a slight change to this section.

[Change] *In situ 3D morphology evolution with quantitative analysis*

Interestingly, the diameter of the porous wire increased significantly when dealloying progressed. This is not seen in other dealloying methods. Considering the thermal expansion, the mean coefficient of thermal expansion ($\bar{\alpha}$) from room temperature to 1000 °F (538 °C) for Ni-20Cr is $8.4 \times 10^{-6}/^\circ\text{F}$ ⁴³. The thermal expansion will only lead to an increase of the diameter by $\sim 0.240 \mu\text{m}$ from the room temperature to 800 °C, which is much smaller than the experimental result. We also noted that a slight sample tilting was observed during the *in situ* experiment, which has been corrected in the visualization; this slight tilted angle could influence the diameter quantification, but was calculated to be $< 1 \mu\text{m}$. **CrCl₃ as a lower X-ray attenuation species was not observed on the surface from the 3D visualizations; in addition, w**We note that the solubility of CrCl_3 in $\text{MgCl}_2\text{-KCl}$ at 800 °C has been measured to be 10.9 mol.%, and thus the Cr should be able to remain as ionic species in the molten salt mixture without saturation given the current experimental conditions; in other words, we do not expect a growth of CrCl_3 layer contributing to the diameter expansion. Hence, the causes for this diameter expansion still require further investigation.

4. Any additional interesting mechanical properties ever found on this material, eq. the sponge-like elasticity enhancement.

[Response] Mechanical properties are indeed of great interest to the community of nanoporous metals and can surely be a scientifically interesting future work. While this work focuses on understanding the kinetics mechanism, we have cited related references in the field, such as the work done by J Weissmüller, K Sieradzki, AM Hodge, ET Lilleodden and other excellent scholars, to highlight the potential future research direction for this material.

[Change] Conclusion

In addition, the heating process could be adjusted to control the coarsening and densification processes, making it possible to tailor the pore size or even to create multi-scale porous structure for a wide range of functional applications. **As studying the mechanical properties has been an active research area for the field, future work can also be conducted to study the mechanical behaviors of the porous metals prepared by molten salt dealloying.**^{9, 23, 78, 79}

5. The XANES spectrum demonstrates the dealloyed structure composed mainly of metallic Ni owing to an absence of surface oxides on the surface after the dealloying process. Were any NiCl_2 compounds if found on the dealloyed structure? A little difference in spectrum between Ni in the dealloyed structure and pure metallic Ni could be related to NiCl_2 but to oxides formation.

[Response] The Ni XANES spectrum of NiCl_2 exhibits a significant white line feature (See Figure R3 below.).⁴ No such feature was present in the XANES spectrum of the sample. We have revised the manuscript accordingly.

Figure R3 Ni K-edge XANES reference compounds spectra.⁴ The characteristic white line feature is indicated with a red arrow

[Change] *Chemical structure study and modeling in molten salt dealloying*

[Change] Both the averaged XANES spectra collected from the central bulk region of the wire [Figure 3 A-B, region of interest 1 (ROI 1)] and from the region near the surface (ROI 2) showed a similar profile and edge position as the ones from a metallic Ni standard, which indicates that the porous structure was mainly metallic Ni after dealloying. Note that the XANES spectra did not exhibit the characteristic white line feature prominent in NiO, NiCl_2 or other NiCl_xO_y compounds.^{60, 61}

Review #3 comments:

This work reports that corrosion of a metal alloy in a molten salt leads to a bicontinuous solid/pore structure in an extended volume of the original solid. The underlying process is therefore analogous to dealloying, as it has previously been described for corrosion in aqueous media are in molten metal. The manuscript analyzes in detail the evolution of the porosity. In-situ synchrotron x-ray tomography is the key technique, supplemented by x-ray near-edge absorption fine structure analysis. Among the findings are insights into the kinetics and the underlying processes of dealloying and of the subsequent coarsening. These insights are of substantial interest in the fields of dealloying as well as molten-salt corrosion. The character of the study, namely bridging communities (scientists interested in making porous metals and those interested in understanding modern-salt corrosion) adds to its relevance. As its key theme, the manuscript proposes molten-salt dealloying as a new mechanism for forming porous materials. This finding, if new, would indeed be a substantial novelty in the field of dealloying. The formation of extended volume of porous material is also of high interest to the corrosion community. Considering those statements, it would appear that the paper is a very suitable submission to Nature Communications.

There is, however, a prior publication by essentially the same team of authors on essentially the same subject, which anticipates the key novelty. I am referring to the manuscripts reference 65, Ronne et al., "Revealing 3D Morphological and Chemical Evolution Mechanisms of Metals in Molten Salt by Multimodal Microscopy", ACS Appl Mater & Interfaces 12 (2020) 17321. Quoting from the abstract of that paper:

"The binary alloy Ni-20Cr developed a bicontinuous porous structure, reassembling functional porous metals manufactured by dealloying. This work elucidates better mechanistic understanding of corrosion in molten salts, which can contribute to the design of more reliable alloys for molten salt applications including next-generation nuclear and solar power plants and opens the possibility of using molten salts to fabricate functional porous materials."

That appears not fundamentally different from the central theme of the present publication. The present publication follows the acting mechanisms to greater depth and thereby does achieve important novel insights which definitely deserve publication. Yet, it would seem that a more specialized journal is suitable for this in-depth follow-up study, rather than Nature Communications with its ambition to report top and qualitatively new insights to a wider audience.

[Response]

We would like to thank the reviewer for the very helpful comments, as well as the positive notion on that the present work *achieved important novel insights*. Below we would like to clarify the relation between the present work and our prior publication in hope that the reviewer may reconsider the merit of the current manuscript.

Our current manuscript marked a substantial advancement compared to our prior work cited by the reviewer. Our prior work focuses on the corrosion of pure Ni vs. Ni binary alloy in the molten salt and the influence of impurities (W). **Percolation dealloying was only hypothesized / proposed as the possible corrosion mechanism** for the Ni alloy. As the reviewer also cited: "The binary alloy Ni-20Cr developed a bicontinuous porous structure, reassembling functional porous metals manufactured by dealloying. This work ... opens the possibility of using molten salts to fabricate functional porous materials." We proposed it based on the observation of one sample that partially developed a bicontinuous structure. Because of the limitation of only analyzing samples after corrosion treatment, we did not observe a complete formation of porous structure. Importantly, the presence of some W impurities in the prior work

also led to a formation of surface shell that hinders a conclusive analysis based on the prior study. **The discussion on the kinetics and mechanisms were entirely lacking in the prior work, and the suggestion on the percolation dealloying remained as a hypothesis.**

In contrast, our current in situ 3D tomography work marks a significant advancement compared to the prior work. Not only was the impurity removed from the system in the current work, the real-time observation of the 3D morphological evolution **proved the percolation dealloying mechanisms in molten salt reaction for the first time and visualized the morphological evolution in 3D in real-time at an elevated temperature, thus enabling quantifying the kinetics on both dealloying and coarsening processes**, revealing the rate-limiting steps and mechanisms.

We would like to emphasize that such achievement is not trivial - this work is **truly the first time that a fully 3D in situ observation was conducted for any dealloying methods where a bicontinuous microstructure formation was directly visualized**. Until now, scientists using either X-ray microscopy or electron microscopy, could only study dealloying processes in situ in 2D, or 3D via post-analysis, thereby lacking the ability to track kinetics in 3D under reaction conditions. Not only just during dealloying (visualization shown in **Figure 2**), **events like the ligament pinch-off and isolated pore formation associated with the coarsening process have been proposed or observed in 2D or post-analysis studies in the field, but were for the first time captured in 3D, in situ by experiments, as shown in our Figure 5**. This significant achievement was made possible thanks to long-haul technical developments, including the construction of a state-of-the art transmission X-ray microscopy beamline offering a world-leading time-resolution for X-ray nano-tomography, design of a miniature in situ heater for the microscope, and finally the development of the in situ sample apparatus presented in this work.

Even considering that we proposed the molten salt dealloying as a potential mechanism in our prior publication, on the notion from Reviewer #3 regarding that such in-depth study should be published in a more specialized journal, we respectfully disagree. We would like to point out, two landmark publications in *Nature Communications* in the field of dealloying in the last 5 years were not about new dealloying methods, but rather about providing significant insights and mechanistic understanding on novel dealloying methods, which is precisely what we are trying to achieve here, in addition to highlighting the new processing method using molten salt dealloying. These two key publications are:

- 1) “Three-dimensional bicontinuous nanoporous materials by vapor phase dealloying” by Lu *et al.*, (2018) *Nature Communications*². The vapor phase dealloying has been demonstrated in 2015 by Sun & Ren, despite the different naming convention: “New preparation method of porous copper powder through vacuum dealloying”⁹.
- 2) “Topology-generating interfacial pattern formation during liquid metal dealloying” by Geslin *et al.* (2015) *Nature Communications*³ This concept of liquid metal dealloying was studied in a pioneer work by Harrison *et al.*¹⁰ in the context of corrosion and only more recently, the method has been further explored as a novel method to fabricate nanoporous materials in 2011.¹¹

The above two papers are influential and have been widely cited in the field, due to their significance in providing kinetics insights despite the fact that they were not the ‘first’ to demonstrate those dealloying methods in the field. We hope that the reviewer could reconsider the merit of our current work – it not only demonstrates for the first time a complete molten salt dealloying (even taking into account our own prior work), but also using a novel characterization method to quantitatively reveal the mechanistic understanding on the materials kinetics.

Detailed comments follow:

1. The manuscript relies overly on the use of abbreviations. This makes the reading arduous and should be corrected.

[Response] The abbreviations used in the manuscript are following:

1) Different dealloying methods including molten salt dealloying (MSD), aqueous solution dealloying (ASD), liquid metal dealloying (LMD) and solid-state interfacial dealloying (SSID); except for the term for molten salt dealloying, these terms have been used in the dealloying literature. However, we will surely reduce the use of abbreviations for the dealloying methods according to the reviewer's comment in the revised manuscript.

2) Synchrotron X-ray methods, including X-ray absorption fine structure (XANES) and transmission X-ray microscopy (TXM). All terms are widely used in the community and thus we will adhere to the convention; however, all terms were properly introduced when they first appear in the manuscript.

3) User facility or beamline names, including National Synchrotron Light Source (NSLS-II), Full-field X-ray Imaging (FXI) beamline, and Beamline for Materials Measurement (BMM). These acronyms are the standard designations for those specific facilities, so we should continue to refer to them that way. All terms were explained when they first appear in the manuscript.

[Change] We changed the acronyms of ASD, LMD and SSID to aqueous solution dealloying, liquid metal dealloying and solid-state interfacial dealloying in the revised manuscript.

2. The discussion of the shift in the Nernst potential addresses graduate-level science that should be well familiar to readers from the dealloying or molten salt corrosion communities; a more concise focus on the novel science might be in order.

[Response] We agree that the discussion may be familiar for the readers from the dealloying or molten salt corrosion communities. We moved this discussion into the Supporting Information as a reference for the broader audience.

[Change] *Kinetics in molten salt dealloying (MSD)*

Thus, the diffusion of oxidants in the molten salt could be one of the rate-limiting factors. On the other hand, for step (iv), the long-range diffusion of the Cr ions produced by dealloying at the alloy-salt interface will also alter the dealloying rate according to Nernst equation (Supporting Information).

On the other hand, for step (iv), the long-range diffusion of the Cr ions produced by dealloying at the alloy-salt interface will also alter the dealloying rate. The long-range diffusion rate influences the local activity (effective concentration) of the Cr ions at the metal-salt interface. Note that the potential for the anodic reaction (E_a), $M \rightarrow M^{n+} + ne^-$, according to Nernst equation is:

$$E_a = E_a^\ominus - \frac{RT}{nF} \ln \frac{a_{M^{n+}}}{a_M}, \quad (7)$$

where E_a^\ominus is the corresponding standard electrode potential, R is the gas constant, and $a_{M^{n+}}$ and a_M are the activity for Cr ions and Cr in the alloy, respectively. When the dealloying electrochemical (anodic) reaction occurs, if Cr ions accumulate locally ($a_{M^{n+}}$ increased) due to a relatively slow diffusion rate, it would cause an increase of the potential for the Cr dissolution reaction and the Gibbs free energy (ΔG) for

the overall corrosion reaction would then become less negative ($\Delta G = -nFE_a$), meaning a decreased driving force for dealloying.

Supporting Information

For step (iv), the long-range diffusion of the Cr ions produced by dealloying at the alloy-salt interface will also alter the dealloying rate. The long-range diffusion rate influences the local activity (effective concentration) of the Cr ions at the metal-salt interface. Note that the potential for the anodic reaction (E_a), $M \rightarrow M^{n+} + ne^-$, according to Nernst equation is:

$$E_a = E_a^0 - \frac{RT}{nF} \ln \frac{a_{M^{n+}}}{a_M}, \quad (1)$$

where E_a^0 is the corresponding standard electrode potential, R is the gas constant, and $a_{M^{n+}}$ and a_M are the activity for Cr ions and Cr in the alloy, respectively. When the dealloying electrochemical (anodic) reaction occurs, if Cr ions accumulate locally ($a_{M^{n+}}$ increased) due to a relatively slow diffusion rate, it would cause an increase of the potential for the Cr dissolution reaction and the Gibbs free energy (ΔG) for the overall corrosion reaction would then become less negative ($\Delta G = -nFE_a$), meaning a decreased driving force for dealloying.

3. The discussion of the state of the art concerning mechanisms for dealloying as well as coarsening misses the insights from atomistic simulations of dealloying that have been documented in the literature of the past five years.

[Response] We thank the reviewer for the comment and have expanded the discussion on the related sections regarding the mechanisms for dealloying and coarsening to further incorporate the insights provided by recent atomistic simulation work of dealloying.

[Change]

Introduction

Research efforts to fundamentally understand the intricate bicontinuous pattern formation^{4, 5, 6, 7, 8, 9} and its coarsening^{10, 11, 12, 13, 14} processes have also provided insights revealing the underlying mechanisms. For instance, continuum simulations (e.g. phase field modeling [^{8, 15, 16}]) and atomistic simulations including kinetic Monte Carlo simulations^{17, 18, 19, 20, 21} and molecular dynamics^{22, 23} model the morphological evolution during both dealloying and coarsening. Simulations also have offered a fundamental understanding on the processing-structure-property relationships such as explaining the atomistic origins on the anomalous compliance²³ and enhanced catalytic properties¹⁷.

Kinetics of Coarsening in Molten Salt

... d^4 vs. t showed the best fit with an R^2 value of 0.9964 which indicated that coarsening in molten salt was dominated by surface diffusion. ... The surface diffusion mechanism is consistent with prior studies on coarsening of nanoporous gold by X-ray nano-tomography¹³ and recent kinetic Monte Carlo simulations⁷³, while the coarsening behavior specifically in a molten salt environment could be further studied via simulations, at both continuum and atomic scales. For instance, the model in the study by

Weissmüller *et al.*⁷³ showed that the degree of surface faceting or roughness has no apparent effect on the coarsening kinetics; such behavior would be of interest to investigate in a molten salt environment as the interfacial energy and reactions differ from the conditions in prior studies.

Another possible mechanism of the isolated pore formation is that neighboring ligaments collapse accompanied by plasticity at nodes or ligaments, supported by prior atomic simulation models.¹⁰ Recent kinetic Monte Carlo and molecular dynamics simulations also revealed that structures with a low relative density exhibited a surface stress-driven dislocation movement leading to coarsening by ligament collapse.²² This mechanism was also further supported by observation via *in situ* transmission electron microscopy in 2D.⁷⁶

4. The data on coarsening kinetics (figure 5) should be displayed in a log-log representation. The discussion of the power-law exponent should include the results of linear regression in that representation. That would provide for an unbiased analysis, strengthening the conclusions.

[Response] We should note that we have done this analysis during the preparation of the manuscript and it arrives to the same conclusion in terms of the coarsening mechanism, as one would expect. Fitting the data via different power-law exponents as used in the manuscript represents that we fit the data with different physical models (different mechanisms)¹², where the model with the best goodness-of-fit would correspond to the most likely mechanism. Both methods are valid, and their results are consistent. We added the figure in a log-log representation and included a linear regression in that representation as advised by the reviewer in Figure 5.

[Change] Figure 5

Figure 5 – Coarsening mechanism power-law fitting for molten salt dealloying (MSD) and a direct observation of enclosed void formation. (A-D) Power-law linear fitting ($n = 1 - 4$) with goodness-of-fit (R^2) value of ligament feature size vs. coarsening time after the microwire has been fully dealloyed (~ 47 min during in situ heating), corresponding to different mechanisms: (A) Viscous flow, (B) evaporation and condensation (here, Ni dissolution and redeposition), (C) bulk diffusion and (D) surface diffusion. **(E) The plot of $\ln[d(t)]$ vs. $\ln(t)$ for measuring the coarsening exponents $n = 1/0.27 = 3.7$ close to 4 indicating**

that surface diffusion is the primary mechanism for coarsening in molten salt. (E-F-F-G) The morphology evolution during the coarsening process at the elevated temperature: (E-F) ligament pinch-off events; the scale bar is 2 μm (F-G) development of an enclosed pore forming within the ligaments.

Kinetics of Coarsening in Molten Salt

The fitting results for the different mechanisms are shown in Figure 5(A-D); d^4 vs. t showed the best fit with an R^2 value of 0.9964 which indicated that coarsening in molten salt was dominated by surface diffusion. The scaling exponent from the plot of $\ln[d(t)]$ vs. $\ln(t)$ was determined to be $1/0.27 = 3.7$, close to 4, indicating that surface diffusion is the primary coarsening mechanism. well (Figure 5 E).

5. Throughout the text, the distinction between 1) external sample volume, 2) volume of the solid phase, and 3) volume of the pore phase, along with 4) solid volume fraction needs a dedicated terminology and precise definitions. These should then be consistently adhered to in the entire manuscript, so that the reader can follow the observations and conclusions on those important parameters clearly. As the text stands, this is not warranted.

[Response] We revised these terms as advised by the reviewer so that the language is consistent when we are referring to the entire sample, the solid phase, or the pore phase.

[Change] *In situ 3D morphology evolution with quantitative analysis*

A representative X-ray projection image and a reconstructed external sample volume rendering are shown in Figure 1(C)

Firstly, the remaining volume of the solid phase is defined as the pixel counts of the solid phase in the dealloyed structure vs. in the pristine wire. The loss volume of the solid phase is then $1 - \text{the remaining volume of the solid phase}$. As shown in Figure 2 (B), the remaining volume of the solid phase Ni-20Cr decreased initially as the dealloying progressed until it reached $\sim 90\%$ after 46 min.

Theoretically, if Cr is completely removed from the Ni-20Cr alloy, the remaining solid-volume fraction of the solid phase would be $\sim 78\%$.

Figure 2 - ... (B-E) Quantitative analysis of the 3D morphological characteristics including the remaining volume and loss volume of the solid phase,

Chemical structure study and modeling in molten salt dealloying

Overall, with the increase of dealloying time, a Ni-rich phase of Ni forms from dealloying Ni-20Cr.

Visualization and quantification analysis of TXM results

A batch segmentation of Ni the solid and pore phases was applied based on the thresholding value between the corresponding peaks in the histogram of the 3D images.

Methods

Visualization of 2D virtual cross-section images and 3D morphology evolution was conducted in commercial software Avizo (Thermo, Fisher Scientific, v9.3) on the whole sample reconstructed data.

The relative remaining volume of the solid phase-loss calculation was determined by the voxel counts of the solid phase in each a given scan divided by the voxel counts of the solid phase in the pristine sample.

6. In several places the paper claims superior friendliness towards the environment. Those passages are not supported by solid arguments. Those arguments should be given (for instance, provide quantifiable and verifiable evidence that dealloying in molten salt is more environmentally friendly than dealloying in acid) or the claims withdrawn.

[Response] We withdrew this claim regarding environmental friendliness in the revised manuscript.

[Change] Introduction

This method avoids using harsh etchants ~~which have been widely used in aqueous solution dealloying and liquid metal dealloying methods~~ and overcomes the oxidation challenge of porous materials. ... In addition, the use of a low-cost and non-toxic salts, and eliminating the need for acids or liquid metals ~~make this method cost-effective and environmentally friendly provide added benefits.~~

Conclusion

MSD ~~can be considered an environmentally friendly and cost-efficient method to create porous metal structures~~ eliminates the use of strong acid ~~to create porous metal structures~~, as in other dealloying methods...

7. For this reviewer's personal taste, the passages Acknowledgment and Author Contributions are way too detailed for what readers are interested in. Scientific papers communicate science – letting them evolve into legally watertight documentation of who did or who financed what is not a trend that scientists should support.

[Response] We revised the Author Contributions section to be more concise.

We believe our acknowledgments section is adequately written to reflect the funding support and collaborators' help we received for the work. The acknowledgements of the funding source and user facilities are required by the funding agencies and user facilities. We also extended our gratitude towards others who helped to make this work possible but were not listed as coauthors.

[Change] Author Contributions

A Ronne and Y-cK Chen-Wiegart developed the research idea with inputs from SM Mahurin and JF Wishart. A Ronne, M Ge, B Layne, S Antonelli, JF Wishart, X Xiao, Chen-Wiegart discussed and designed the *in situ* X-ray nano-tomography experiments. P Halstenberg purified the salts under the supervision of S Dai. D. S. Maltsev and A. S. Ivanov conducted the solubility measurements. ~~B Layne and A Ronne designed the double-capillary *in situ* sample scheme. The details of the double-capillary setup, design were primarily developed and tested by B Layne and A Ronne, with inputs from X Xiao, M Ge, JF Wishart and Y-cK Chen-Wiegart. S Antonelli and X Xiao designed and tested the heater, with inputs from W-K Lee, M Ge, B Layne, JF Wishart and Y-cK Chen-Wiegart. W-K Lee, M Ge and X Xiao designed, commissioned and set up the FXI beamline.~~ A Ronne, X Liu, L-C Yu and Y-cK Chen-Wiegart conducted the FXI beamtime with the support of X Xiao, M Ge, W-K Lee and Chen-Wiegart group members. ~~X Xiao developed the programs for tomographic reconstruction. A Ronne reconstructed the tomographic data with the advice from X Xiao and the support from the Chen-Wiegart group members. The concept of the image data analysis and algorithms were developed and implemented by X Liu with the inputs and assistance from C-C Lin, A Ronne, L-C Yu and Y-cK Chen-Wiegart. The 3D data visualization was carried out by L-C Yu, with inputs from X Liu, A Ronne, and Y-cK Chen-Wiegart.~~ X Liu, A Ronne and L-C Yu conducted data analysis and visualization under the guidance of X Xiao and Y-cK Chen-Wiegart. XANES modeling was conducted by Y Liu and A. I. Frenkel. *In situ* videos were

prepared by X Liu. ~~Figures were prepared by X Liu, A Ronne, and L Yu.~~ X Liu, A Ronne, and Y-cK Chen-Wiegart developed the data interpretation and mechanistic understanding and wrote the manuscript with inputs from all co-authors.

References:

1. Liu P, *et al.* Dealloying Kinetics of AgAu Nanoparticles by In Situ Liquid-Cell Scanning Transmission Electron Microscopy. *Nano Letters* **20**, 1944-1951 (2020).
2. Lu Z, *et al.* Three-dimensional bicontinuous nanoporous materials by vapor phase dealloying. *Nature Communications* **9**, (2018).
3. Geslin PA, McCue I, Gaskey B, Erlebacher J, Karma A. Topology-generating interfacial pattern formation during liquid metal dealloying. *Nature Communications* **6**, (2015).
4. Van Loon LL, Throssell C, Dutton MD. Comparison of nickel speciation in workplace aerosol samples using sequential extraction analysis and X-ray absorption near-edge structure spectroscopy. *Environmental Science-Processes & Impacts* **17**, 922-931 (2015).
5. Tian Y, *et al.* Speciation of nickel (II) chloride complexes in hydrothermal fluids: In situ XAS study. *Chemical Geology* **334**, 345-363 (2012).
6. Jin K, *et al.* Thermophysical properties of Ni-containing single-phase concentrated solid solution alloys. *Materials & Design* **117**, 185-192 (2017).
7. Bell S, Steinberg T, Will G. Corrosion mechanisms in molten salt thermal energy storage for concentrating solar power. *Renewable & Sustainable Energy Reviews* **114**, (2019).
8. Guo SQ, Zhang JS, Wu W, Zhou WT. Corrosion in the molten fluoride and chloride salts and materials development for nuclear applications. *Progress in Materials Science* **97**, 448-487 (2018).
9. Sun YX, Ren YB. New preparation method of porous copper powder through vacuum dealloying. *Vacuum* **122**, 215-217 (2015).
10. Harrison JD, Wagner C. THE ATTACK OF SOLID ALLOYS BY LIQUID METALS AND SALT MELTS. *Acta Metallurgica* **7**, 722-735 (1959).
11. Wada T, Setyawan AD, Yubuta K, Kato H. Nano- to submicro-porous beta-Ti alloy prepared from dealloying in a metallic melt. *Scripta Materialia* **65**, 532-535 (2011).

12. Andrews WB, Elder KLM, Voorhees PW, Thornton K. Effect of transport mechanism on the coarsening of bicontinuous structures: A comparison between bulk and surface diffusion. *Physical Review Materials* **4**, (2020).

REVIEWERS' COMMENTS

Reviewer #1 (Remarks to the Author):

The authors properly responded this reviewer's comments. The revised manuscript clarifies the accomplishments and limitation in an articulate manner. Thus, the manuscript should be accepted for publication.

Reviewer #2 (Remarks to the Author):

This paper should be accepted for publication.

Reviewer #3 (Remarks to the Author):

The authors have satisfactorily replied to my questions. Specifically they provide valid statements on the novelty of the study, even in view of their prior publication. I can now recommend the manuscript for publication.

Responses to the reviewers

Reviewer #1

The authors properly responded this reviewer's comments. The revised manuscript clarifies the accomplishments and limitation in an articulate manner. Thus, the manuscript should be accepted for publication.

[Response] We would like to thank Reviewer #1 again for the insightful comments helping to revise and strengthen the manuscript.

Reviewer #2

This paper should be accepted for publication.

[Response] We appreciate all the helpful insights and comments from Reviewer #2.

Reviewer #3

The authors have satisfactorily replied to my questions. Specifically they provide valid statements on the novelty of the study, even in view of their prior publication. I can now recommend the manuscript for publication.

[Response] We are grateful for the insights and comments from Reviewer #3, which helped us to improve the manuscript.